# DEEP ECG-REPORT INTERACTION FRAMEWORK FOR CROSS-MODAL REPRESENTATION LEARNING

## ABSTRACT

Electrocardiogram (ECG) is of great importance for the clinical diagnosis of cardiac conditions. Although existing self-supervised learning methods have obtained great performance on learning representation for ECG-based cardiac conditions classification, the clinical semantics can not be effectively captured. To overcome this limitation, we proposed a **D**eep **ECG-R**eport **I**nteraction (**DERI**) framework to learn cross-modal representations that contain more clinical semantics. Specifically, we design a novel framework combining multiple alignments and mutual feature reconstructions to learn effective cross-modal representation of the ECG-Report, which fuses the clinical semantics of the report into the learned representation. An RME-Module inspired by masked modeling is proposed to improve the ECG representation learning. Furthermore, we extend ECG representation learning with a language model to report generation, which is significant for evaluating clinical semantics in the learned representations and even clinical applications. Comprehensive experiments on various datasets with various experimental settings show the superior performance of our proposed DERI.

## 1 INTRODUCTION

Electrocardiogram (ECG) is a widely used data for reflecting heart electrical activity (Attia et al. (2019); Al-Zaiti et al. (2023)), which is of great importance in cardiac conditions classification. Supervised learning methods have obtained effective performance in ECG signal classification with high-quality annotations (Huang et al. (2022); Chen et al. (2024a)). However, there are a large number of unlabeled ECG signals in the real world, and supervised learning methods have difficulty utilizing this resource effectively. To reduce the dependence on labeled data, ECG representation learning methods based on self-supervised learning demonstrate their powerful performance (Oh et al. (2022); Mehari & Strodthoff (2022); Lan et al. (2022); Liu et al. (2024c)). Compared to supervised learning methods, self-supervised learning methods aim to learn effective representations from ECG signal data in a detached labeling context, and thus tend to be more generalizable and adaptable to different downstream tasks, showing great potential.

Existing self-supervised methods for ECG signals can be broadly categorized into generative learning and contrastive learning approaches. Generative learning focuses on masked modeling to reconstruct or generate the input data itself, while contrastive learning aims to learn more discriminative representations by comparing similar and dissimilar samples. However, most of these methods are single-modal, limiting their ability to effectively capture comprehensive, deep semantic representations of ECG signals. Specifically, generative learning methods, which emphasize signal recovery through masked modeling, primarily focus on low-level signal patterns (e.g., local intensities and waveforms), often overlooking high-level semantic features related to clinical conditions (Zhang et al. (2023); Liu et al. (2023b)). Contrastive approaches, on the other hand, typically create augmented views as positive pairs by applying data transformations at the input level. However, these transformations can distort the inherent semantics of the ECG signal, leading to a loss of critical information (Lan et al. (2024); Na et al. (2024)). Multi-modal learning has emerged as a promising solution for these limitations due to its ability to learn effective representations from multiple data sources (Chen et al. (2024b;c)). Compared to ECG signals, clinical reports offer direct high-level semantic insights. Inspired by advances in medical imaging and radiology reports (Chen et al. (2023); Liu et al. (2023a)), Liu et al. proposed a multi-modal representation learning approach by aligning ECG signals with clinical reports (Liu et al. (2024b)). However, their method aligns ECG features

with report features in the feature space, drawing inspiration from CLIP (Radford et al. (2021)), but the interaction between modalities is relatively shallow. Furthermore, although the ECG representations learned by MERL perform well in classification tasks, they fail to effectively convey the underlying semantics of ECG recordings, which are crucial for understanding cardiac conditions.

To overcome these limitations, we proposed a novel **D**eep **E**CG **R**eport **I**nteraction (**DERI**) framework for cross-modal representation learning. To better capture the clinical semantics of ECG signals and reports, we design an encoder-decoder structure to conduct multiple alignments and feature reconstruction. Specifically, the ECG signals and their corresponding clinical reports are first encoded and projected into a shared alignment space to achieve an initial alignment. To enhance interaction, two specialized decoders are employed to reconstruct features by decoding the aligned representations into the encoding of the other modality. This reconstruction process captures the latent structures in both modalities, enabling the learning of richer cross-modal representations. Subsequently, the decoded features are fused with the modality-specific aligned features to create mixed representations incorporating both ECG signal and clinical report semantics. These mixed representations are further utilized for a second alignment. Additionally, we introduce a Random Masked Enhancement Module (RME-Module) to improve ECG representation learning. Furthermore, the proposed DERI model is integrated with language models to generate clinical reports, providing a way to assess the learned clinical semantics embedded in the representations. Extensive experiments across various settings and datasets are conducted to demonstrate the effectiveness of the proposed approach. The main contributions of this work are summarized as follows:

- To learn effective ECG representation for cardiac conditions from reports, we propose a novel cross-modal framework of ECG-Report via multiple feature alignment and mutual feature reconstruction. An RME-Module is also designed for ECG representation learning enhancement.
- To better illustrate the clinical semantics learned by DERI, we combine it with a language model for report generation. The pre-trained model provides effective ECG representation and a language model is used to decode it into clinical reports, which can provide clinical semantics visually.
- Comprehensive experiments on downstream datasets are conducted to evaluate the proposed DERI method, including zero-shot classification, linear probing, and even report generation. Experimental results illustrate that our proposed DERI method surpasses all SOTA methods by a large margin, which represents that our method can learn more effective clinical semantics.

## 2 RELATED WORK

### 2.1 SINGLE-MODAL ECG REPRESENTATION LEARNING

There are various self-supervised learning methods for ECG representation learning. Most of these methods are single-modal, which conduct generative learning or contrastive learning on unannotated ECG signals. CLOCS (Kiyasseh et al. (2021)) and ASTCL (Wang et al. (2023)) are the SOTA single-modal contrastive learning methods that explore temporal and spatial correlations of ECG signals. Similarly, ST-MEM (Na et al. (2024)) proposes to learn ECG representation by spatial-temporal masking modeling and reconstruction of 12-lead ECG signals. Although all these unimodal methods have achieved good performance, they still fall short in learning the clinical semantics of ECG signals (Liu et al. (2024b)). Single-modal contrastive and generative methods extract representations only from ECG signals and are therefore not related to diagnostic reports.

### 2.2 MULTI-MODAL ECG REPRESENTATION LEARNING

Several works conduct ECG multi-modal learning for better classification. Raghu et al. (2022) proposes to learn representations from ECG signals and structured data from labs and vitals by contrastive learning. Lalam et al. (2023) combines ECG signals with structured Electronic Health Records (EHRs) to conduct contrastive learning. BPNet fuses ECG signals with PPG signals to better conduct blood pressure estimation (Long & Wang (2023)). However, these methods do not use diagnostic report data, making it difficult to learn the clinical semantics effectively. To learn the clinical semantics of ECG signals, Liu et al. proposed to align ECG features with clinical reports inspired by multi-modal learning in medical images and radiology reports (Liu et al. (2024a;b)). Intro-

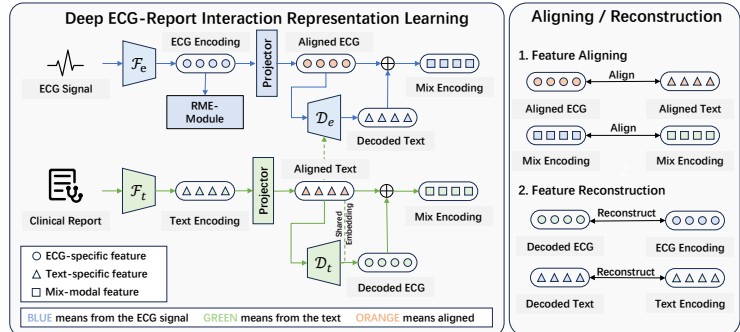

Figure 1: Framework of proposed DERI for ECG-Report multi-modal representation learning.

ducing corresponding diagnostic reports for ECG representation learning greatly improves their performance on the downstream cardiac condition classification task, but these multimodal approaches only achieve shallow modal interaction. The learned representations are also not able to efficiently incorporate the semantics in the reports. Therefore, we design our proposed DERI method to conduct deep cross-modal interaction and then expand the model to report generation, which is of great meaning for clinical diagnosis.

## 2.3 CLINICAL REPORT GENERATION

Clinical report generation in radiology has obtained great performance inspired by imaging captioning You et al. (2016; 2021). R2Gen (Chen et al. (2020b)) uses a memory-driven Transformer to generate a radiology report directly with the representation of the medical image. CvT2DistilGPT2 (Liu et al. (2023b)) demonstrates that pre-trained NLP models can provide benefits for radiology report generation as well. X-REM proposed to fuse the image-text multi-modal representation and then used retrieval-based methods to generate the predicted reports from the retrieval corpus (Jeong et al. (2024)). Inspired by these image-based clinical report generation methods, we extend our proposed ECG representation learning method to ECG-based clinical report generation, which can help understand the clinical semantics of the cardiac conditions from the ECG signals.

## 3 METHODOLOGY

### 3.1 OVERVIEW

Our DERI framework is designed to learn effective multi-modal representation from ECG signals and corresponding clinical reports. Compared to learning ECG representation alone, our DERI can obtain much deeper interaction between signal modal and text modal via feature aligning and reconstruction. As depicted in Figure 1, we adopted a dual-encoder framework as the backbone. Our DERI performs modal interaction representation learning from feature alignment and feature reconstruction. We also design an RME-Module to enhance the ECG representation learning. Deeper modal interactions enable learned representations to include more clinical information, from satisfying cardiac status classification to generating diagnostic reports.

### 3.2 ECG-REPORT MULTIPLE ALIGNMENT

The ECG-Report Alignment in DERI contains two strategies including modal-specify feature alignment and mix-modal feature alignment. Given an ECG signal recording $e_i$ with corresponding clinical report $r_i$, we construct an ECG-Report pair as $(e_i, r_i)$, with $i = 1, 2, 3, ..., N$ where $N$ is the number of recordings. Two distinct encoders $\mathcal{F}_e$ and $\mathcal{F}_t$ are used to learn the latent encoding of ECG signal and report text respectively, represented as $z_{e,i}$ and $z_{t,i}$. Specifically, the latent encoding is obtained by $z_{e,i} = \mathcal{F}_e(e_i)$ and $z_{t,i} = \mathcal{F}_t(r_i)$. To align the ECG encoding and text encoding, we use two linear projectors $\mathcal{P}_e$ and $\mathcal{P}_t$ to map them into an alignment space of the same dimension, which can be represented as $\hat{A}_{e,i} = \mathcal{P}_e(z_{e,i})$ and $\hat{A}_{t,i} = \mathcal{P}_t(z_{t,i})$. The align loss $\mathcal{L}_{align}$, which is inspired by the CLIP loss (Radford et al. (2021)), to close the distance between the representations

of ECG and report in the alignment space. Specifically, each ECG recording and the corresponding report are regarded as a positive pair. The loss function $\mathcal{L}_{align}$ is expressed as Eq. 1 to Eq. 3.

$$\mathcal{L}_{align} = \frac{1}{2B} \sum_{i=1}^{N} \sum_{j=1}^{N} (\mathcal{L}_{i,j}^{e,t} + \mathcal{L}_{i,j}^{t,e}), \qquad (1)$$

$$\mathcal{L}_{i,j}^{x,y} = -log \frac{exp(s_{i,j}^{x,y}/\tau)}{\sum_{k=1}^{B} \mathbb{I}_{[k \neq i]} exp(s_{i,j}^{x,y}/\tau)}, x \in [e,t], y \in [e,t], \qquad (2)$$

$$s_{i,j}^{x,y} = \frac{\hat{A}_{x,i}^{\top} \hat{A}_{y,j}}{\| \hat{A}_{x,i} \| \| \hat{A}_{y,j} \|}, \qquad (3)$$

where $B$ is the batch size, $x \in [e,t], y \in [e,t]$ represents the ECG modal or the text modal, $\tau$ is the temperature coefficient which is set as 0.7, $\mathbb{I}$ is the indicator function. Through calculating the align losses of ECG-report $\mathcal{L}_{i,j}^{e,t}$ and report-ECG $\mathcal{L}_{i,j}^{t,e}$, the model can perform better feature alignment.

Furthermore, to conduct deep ECG-report interaction, we adopt Cross-Modal Reconstruction (depicted in Section 3.3) to decode the representation of another mode ($D_e$ and $D_t$ represent decoded ECG feature and decoded text feature) from the aligned modal representation. Then the decoded features are added to the aligned modal representation to obtain the mix-modal encoding. By combining alignment and reconstruction, mixed-modal can better achieve deep ECG-report interaction. Specifically, we will use the $\hat{A}_{e,i}$ and $\hat{A}_{t,i}$ as the core to obtain the mix-modal encoding $\hat{M}_i^e = \hat{A}_{e,i} \oplus D_t$ and $\hat{M}_i^t = \hat{A}_{t,i} \oplus D_e$, and then conduct encoding alignment in mixed space, thus obtaining the final multimodal representation. The mixed alignment loss can be calculated as Eq. 4.

$$\mathcal{L}_{mixed} = \frac{1}{2B} \sum_{i=1}^{N} \sum_{j=1}^{N} (\mathcal{L'}_{i,j}^{e,t} + \mathcal{L'}_{i,j}^{t,e}), \qquad (4)$$

where $\mathcal{L'}_{i,j}^{e,t}$ and $\mathcal{L'}_{i,j}^{t,e}$ is calculated by using $\hat{M}_i^e$ and $\hat{M}_i^t$ to replace $\hat{A}_{e,i}$ and $\hat{A}_{t,i}$ as Eq. 2 and Eq. 3. Therefore, our proposed method can effectively extract mixed modal representations with report characteristics by using only ECG signals after completing the pre-training stage, and can better complete the task of zero sample classification and report generation.

In conclusion, the whole loss for multiple ECG-report alignment can be written as Eq. 5.

$$\mathcal{L}_{ERA} = \mathcal{L}_{align} + \mathcal{L}_{mixed} \qquad (5)$$

### 3.3 CROSS-MODAL MUTUAL RECONSTRUCTION

To better guide the model in achieving deeper modal interactions between ECG signals and diagnostic reports, we introduce cross-modal mutual reconstruction. Specifically, after we obtained the aligned ECG feature $\hat{A}_{e,i}$ and the aligned Text features $\hat{A}_{t,i}$, we aim to facilitate modal interactions by reconstructing the target modes while bringing them closer to each other in space. We introduce decoder transformers to decode the semantic space of one modal in the alignment space to another modal. Reports offer intuitive semantic information valuable for heart state classification but are often unavailable. Therefore, we introduce a shared embedding derived from the textual modality decoder. This shared embedding is combined with the ECG features, enriching them with additional textual features to enhance heart state classification. After completing the pre-training in this manner, the final representation obtained from inputting only the ECG data effectively encapsulates the semantic information of the corresponding report text. This process is represented as: Eq. 6.

$$\hat{D}_{e,i} = \mathcal{D}_e(\hat{A}_{t,i}), \;\; \hat{D}_{t,i} = \mathcal{D}_t(Concat[\hat{A}_{e,i}, SE_t]), \qquad (6)$$

where $\mathcal{D}_e$ and $\mathcal{D}_t$ are the decoder transformers to obtain ECG encoding $\hat{D}_{e,i}$ and report encoding $\hat{D}_{t,i}$ respectively, and $SE_t$ are the shared embedding. Then we use standard contrastive loss on the original feature embeddings and the decoded embeddings for cross-modal reconstruction as Eq. 7.

$$\mathcal{L}_{CMR} = \mathcal{L}_{DEC}^e + \mathcal{L}_{DEC}^t = \frac{1}{2B} \sum_{i=1}^{N} \sum_{j=1}^{N} (\mathcal{L}_{i,j}^{ze,de} + \mathcal{L}_{i,j}^{zt,dt}) \qquad (7)$$

Figure 2: Pipeline of zero-shot classification and report generation of our proposed DERI.

where $\mathcal{L}_{DEC}^e$ and $\mathcal{L}_{DEC}^t$ represents the loss of ECG and report feature reconstruction respectively, $\mathcal{L}_{i,j}^{ze,de}$ and $\mathcal{L}_{i,j}^{zt,dt}$ represent to use $\mathcal{D}_e$ and $\mathcal{D}_t$ with the original features $z_{e,i}$ and $z_{t,i}$ to calculate the similarity as the same of Eq. 3.

### 3.4 LATENT RANDOM MASKING ENHANCEMENT

We further conduct an RME module on latent ECG encoding to facilitate representation learning. Considering that augmentation directly at the data level entails the loss of semantic information about the signal, we use our proposed RME-Module on the ECG encoding. Specifically, as the Encoder extracts encodings, it tends to focus on the local features of the signal to form an encoding sequence. Rather than applying global average pooling to the ECG encoding sequence and using two separate dropout operations to create augmented views, we instead randomly mask the sequence twice, independently. Then, a multi-head attention mechanism is employed to aggregate the sequence, producing two augmented views of the encoding as a positive pair. This random masking approach helps preserve sequence-level semantic features while enabling the model to learn global features more effectively. We then use standard contrastive loss on these two augmented encoding views. The whole process can be illustrated in Eq. 8

$$\mathcal{L}_{RME} = -\frac{1}{L} \sum_{i=1}^{N} \sum_{j=1}^{N} log \frac{exp(s_{i,j}\tau)}{\sum_{k=1}^{L} \mathbb{I}_{[k \neq i]} exp(s_{i,j}^{x,y}/\tau)},$$
$$\text{where } s_{i,i} = z_{e,i}^{1\top} z_{e,i}^2, \tag{8}$$
$$z_{e,i}^1 = MHA(\text{Mask}(z_{e,i})) = MHA(\mathcal{M}_1 \times z_{e,i}),$$
$$z_{e,i}^2 = MHA(\text{Mask}(z_{e,i})) = MHA(\mathcal{M}_2 \times z_{e,i}),$$

where $MHA$ is a multi-head attention, Mask is the random mask strategy, which generates random mask sets $\mathcal{M}_1$ and $\mathcal{M}_2$. $\mathcal{M}_1$ and $\mathcal{M}_2$ with each entry independently sampled with masking ratio $p = 0.1$ are in $\mathbb{R}^{b \times n}$ where $b$ is the batch size and $n$ is the length of the embedding sequence. Each item in $\mathcal{M}$ is either 0 or 1, indicating whether the corresponding patch should be masked. We add a global average on the $MHA$ to obtain the global representation of the masked embedding. Importantly, the random masks are generated by two independent random noises.

In summary, our proposed DERI learns representative ECG features with the help of clinical reports by jointly minimizing $\mathcal{L}_{ERA}$, $\mathcal{L}_{CMR}$ and $\mathcal{L}_{RME}$. The overall training loss can shown as Eq. 9.

$$\mathcal{L}_{total} = \mathcal{L}_{ERA} + \mathcal{L}_{CMR} + \mathcal{L}_{RME} \tag{9}$$

### 3.5 DOWNSTREAM TASKS ON DERI FRAMEWORK

After training the proposed DERI model, we can obtain an effective representation of ECG signals that contains clinical report information. Then we can use the representation to conduct zero-shot classification and report generation. Considering the quality of the category prompts for zero-shot classification will have a great impact on the performance (Pratt et al. (2023); Maniparambil et al. (2023)), we adopt the CKEPE prompts which are constructed by combining large language model

(LLM) and clinical knowledge (Liu et al. (2024b)). The whole process of these two downstream tasks is illustrated in Figure 2.

**Zero-shot Classification.** We adopted CKEPE (Liu et al. (2024b)) as the category prompts and used the trained report encoder $\mathcal{F}_t$ and the projector to obtain the prompt embeddings of all categories. We then use the trained DERI model to obtain the Mix Encoding, which is the final representation containing both ECG signal features and the corresponding clinical report features. Finally, we calculate the similarity between the Mix Encoding and the prompt encoding and then adopt the category to which the prompt with the highest similarity belongs as the classification result. Importantly, all the parameters of the proposed DERI are frozen in this process.

**Report Generation.** After we obtain the final representation of ECG, we adopt GPT-2 as the text decoder to construct an encoder-decoder structure since DistilGPT2 (Sanh (2019)) has shown its great performance on radiology report generation (Nicolson et al. (2023); Wang et al. (2024)). We adopt a trainable linear layer to transform the dimension of the input encoding to meet the dimension of the GPT-2 and conduct fine-tuning on the GPT-2 to minimize a cross-entropy loss $\mathcal{L}_{CE}$ between the generated report to the ground truth reports. After training the linear layer and fine-tuning the GPT-2, we can generate corresponding diagnostic reports with ECG signals alone.

# 4 EXPERIMENT

## 4.1 DATASETS

**MIMIC-ECG** The proposed DERI model is pre-trained on the MIMIC-ECG dataset (Gow et al. (2023)), which contains 800,035 paired ECG signals and clinical reports from 161,352 unique subjects. We removed all the samples without reports containing more than 3 words and replaced 'NaN' or 'Inf' in the ECG signal with mean interpolation. Finally, the dataset used for pre-training has 771,693 samples. We also conduct report generation tasks on this dataset.

**PTBXL** (Wagner et al. (2020)) contains 21,837 12-lead ECG signals recorded from 18,885 patients at a sample rate of 500 Hz with a duration of 10 seconds. It contains four multi-label classification tasks: Superclass, Subclass, Form, and Rhythm. The four different tasks are divided based on the ECG annotation protocol. These four tasks are multi-label classification tasks.

**CPSC2018** (Liu et al. (2018)) is a publicly accessible dataset that contains 6,877 12-lead ECG recordings with a sampling rate of 500 Hz. The duration of these signals ranges from 6 to 60 seconds. Each recording has one corresponding label within nine categories.

**Chapman-Shaoxing-Ningbo (CSN)** (Zheng et al. (2020; 2022)) is a publicly accessible dataset that contains 45,152 12-lead ECG recordings with a sampling rate of 500 Hz. Each recording has a duration of 10 seconds and the ECG signals with "unknown" annotation are removed. Therefore, 23,026 ECG records with 38 categories were used for classification.

## 4.2 EXPERIMENTAL SETUP

**Pre-training.** For the encoders used for ECG and reports, we adopt a random initialized 1D-ResNet18 and the Med-CPT (Jin et al. (2023)) respectively. For decoders, we adopt two transformers with 8 attention heads, a depth of 2, and a hidden size of 256 respectively for ecg encoding reconstruction and report encoding reconstruction. We use the AdamW optimizer with a learning rate of 1e-3 and a weight decay of 1e-8. The epoch for pre-training is set as 50 with a cosine annealing scheduler to adjust the learning rate. We conduct all the pre-trained experiments on four NVIDIA GeForce RTX 4090 GPUs with a batch size of 512 per GPU.

**Classification.** We freeze the whole DERI and conduct zero-shot classification as illustrated in Section 3.5. For linear probing, we add a new linear classifier and freeze all other parameters in our DERI. We adopt three different settings, which utilize 1%, 10%, and 100% of the training data to train the linear classifier. Since these tasks are all classifications that contain many categories, we adopt the macro AUC as the evaluated metric. We conduct these experiments on one NVIDIA GeForce RTX 4090 GPU and more details about the implementation are provided in the Appendix B. The baselines we compared include single-modal methods such as **ASTCL.** (Wang et al. (2023)),

**CRT** (Zhang et al. (2023)), **STMEM** (Na et al. (2024)) and multi-modal method MERL. More details can be found in the Appendix C.

**Report generation.** We adopt the Natural Language Generation (NLG) metrics to evaluate our report generation performance first, which include BLEU-n for n-gram overlap evaluation (Papineni et al. (2002)) and ROUGE-L with the longest common sub-sequence between the original report and generated report (Lin (2004)). Considering that NLG metrics can not effectively reflect the clinical accuracy of the generated report, we further integrated Clinical Efficiency (CE) metrics inspired by the zero-shot classification (more details can be found in Appendix E). Specifically, we use the pre-trained text decoder to obtain prompt embeddings of all categories in the CKEPE. We feed the pre-trained text decoder with the ground clinical report and calculate the similarity as the classification probability for the category. We adopt the category with the highest classification probability as the ground truth label. Then we use the generated report to obtain the predicted label in the same way. The predicted labels are then used to compute precision, recall, and F1 scores against ground truths as the CE metrics. We conduct the experiments on two NVIDIA GeForce RTX 4090 GPUs.

### 4.3 RESULTS ON CLASSIFICATION

Since most of the existing ECG representation learning methods are proposed without a text encoder for zero-shot learning, we compared our proposed DERI with MERL to verify the cross-modal ECG representations learned from clinical reports. The comparison results are illustrated in Figure 3. It is evident that our proposed method, DERI, significantly outperforms MERL across all tasks. The dotted line in the figure indicates the average performance of the two zero-sample methods on the six classification tasks. DERI achieves an average macro AUC of 78.73, while MERL attains only 75.25. This underscores DERI's capability to learn clinically relevant representations through deep cross-modal interactions between ECG signals and diagnostic reports.

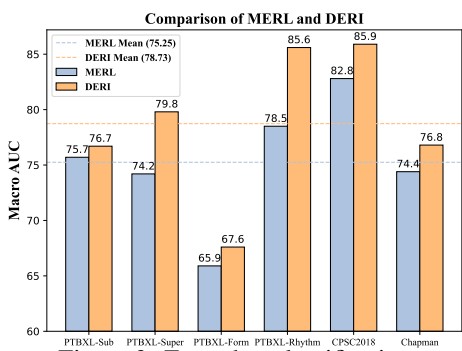

Figure 3: Zero-shot classification

Compared to MERL which just aligns the ECG and report encoding, our proposed DERI achieves deep cross-modal interaction by multiple alignment and feature reconstruction, which enables the model to learn more effective representation for zero-shot clinical classification.

To better evaluate the performance of our proposed DERI on ECG representation learning, we conduct linear probing and compare the result with other ECG self-supervised learning methods because linear probing has been a standardized procedure for self-supervised learning (Wang et al. (2023)). The experimental results are shown in Table 1. Random Init refers to using the proposed DERI framework to obtain the ECG-specific mixed encoding without pre-training, followed by training the model in a supervised learning setting for classification. It can be observed that DERI outperforms all the baselines including multi-modal method MERL and other single-modal self-supervised learning methods across all settings of training data ratio and datasets. Interestingly we found that DERI achieved the greatest performance advantage when using only 1% of the training data.

Table 1: Linear probing AUC. We bold the best results and grey represents the second highest.

| Method | PTBXL-Super | | | PTBXL-Sub | | | PTBXL-Form | | | PTBXL-Rhythm | | | CPSC2018 | | | CSN | | |
|---|---|---|---|---|---|---|---|---|---|---|---|---|---|---|---|---|---|---|
| | 1% | 10% | 100% | 1% | 10% | 100% | 1% | 10% | 100% | 1% | 10% | 100% | 1% | 10% | 100% | 1% | 10% | 100% |
| SimCLR | 63.41 | 69.77 | 73.53 | 60.84 | 68.27 | 73.39 | 54.98 | 56.97 | 62.52 | 51.41 | 69.44 | 77.73 | 59.78 | 68.52 | 76.54 | 59.02 | 67.26 | 73.20 |
| BYOL | 71.70 | 73.83 | 76.45 | 57.16 | 67.44 | 71.64 | 48.73 | 61.63 | 70.82 | 41.99 | 74.40 | 77.17 | 60.88 | 74.42 | 78.75 | 54.20 | 71.92 | 74.69 |
| BarlowTwins | 72.87 | 75.96 | 78.41 | 62.57 | 70.84 | 74.34 | 52.12 | 60.39 | 66.14 | 50.12 | 73.54 | 77.62 | 55.12 | 72.75 | 78.39 | 60.72 | 71.64 | 77.43 |
| MoCo-v3 | 73.19 | 76.65 | 78.26 | 55.88 | 69.21 | 76.69 | 50.32 | 63.71 | 71.31 | 51.38 | 71.66 | 74.33 | 62.13 | 76.74 | 75.29 | 54.61 | 74.26 | 77.68 |
| SimSiam | 73.15 | 72.70 | 75.63 | 62.52 | 69.31 | 76.38 | 55.16 | 62.91 | 71.31 | 49.30 | 69.47 | 75.92 | 58.35 | 72.89 | 75.31 | 58.25 | 68.61 | 77.41 |
| TS-TCC | 70.73 | 75.88 | 78.91 | 53.54 | 66.98 | 77.87 | 48.04 | 61.79 | 71.18 | 43.34 | 69.48 | 78.23 | 57.07 | 73.62 | 78.72 | 55.26 | 68.48 | 76.79 |
| CLOCS | 68.94 | 73.36 | 76.31 | 57.94 | 72.55 | 76.24 | 51.97 | 57.96 | 72.65 | 47.19 | 71.88 | 76.31 | 59.59 | 77.78 | 77.49 | 54.38 | 71.93 | 76.13 |
| ASTCL | 72.51 | 77.31 | 81.02 | 61.86 | 68.77 | 76.51 | 44.14 | 60.93 | 66.99 | 52.38 | 71.98 | 76.05 | 57.90 | 77.01 | 79.51 | 56.40 | 70.87 | 75.79 |
| CRT | 69.68 | 78.24 | 77.24 | 61.98 | 70.82 | 78.67 | 46.41 | 59.49 | 68.73 | 47.44 | 73.52 | 74.41 | 58.01 | 76.43 | 82.03 | 56.21 | 73.70 | 78.80 |
| STMEM | 61.12 | 66.87 | 71.36 | 54.12 | 57.86 | 63.59 | 55.71 | 59.99 | 66.07 | 51.12 | 65.44 | 74.85 | 56.69 | 63.32 | 70.39 | 59.77 | 66.87 | 71.36 |
| MERL | 82.39 | 86.27 | 88.67 | 64.90 | 80.56 | 84.72 | 58.26 | 72.43 | 79.65 | 53.33 | 82.88 | 88.34 | 70.33 | 85.32 | 90.57 | 66.60 | 82.74 | 87.95 |
| Random Init | 78.10 | 84.33 | 89.47 | 69.96 | 78.70 | 84.01 | 60.82 | 67.14 | 70.08 | 60.12 | 79.34 | 83.98 | 64.67 | 75.57 | 91.45 | 67.42 | 74.54 | 79.97 |
| DERI | 85.46 | 89.84 | 90.52 | 73.50 | 80.60 | 85.52 | 62.53 | 72.51 | 84.37 | 65.44 | 83.66 | 92.34 | 79.45 | 89.40 | 93.45 | 77.93 | 87.86 | 91.93 |

Notably, on the PTBXL-Super task, our proposed DERI with 1% training data outperforms all the single-modal ingle-modal self-supervised learning methods with 100% training data, just below the multi-modal method MERL. We can also observe that the multimodal methods MERL and DERI

show excellent performance on all classification tasks with the training data setup, reflecting the effectiveness of ECG representation learning in conjunction with clinical reports. The same significant performance advantage of DERI over MERL also exists, which implies that our proposed method is more effective in realizing cross-modal representation learning of ECG with clinical reports to extract important clinical information for cardiac condition classification.

Distribution shift can effectively validate the learning ability of representation learning models for different data domains, to evaluate their generalizability and robustness. Therefore, we adopt linear probing with 100% training data to re-train the classification heads on three classification tasks, namely, PTBXL-Super, CPSC2018, and CSN, to conduct distribution shift experiments. It is worth noting that since both the MERL and DERI methods achieve zero-sample classification, we do not need to re-train the classification head for the model, but only need to reclassify the zero-sample classification results. Specifically, we trained the single-modal supervised learning methods with linear probing on one dataset (source domain) and then tested it on other datasets (target domain).

Table 2: Distribution shift AUC. We bold the best results and grey represents the second highest.

| Source Domain Target Domain | Zero-shot | Training Data Ratio | PTBXL-Super CPSC2018 | CSN | CPSC2018 PTBXL-Super | CSN | CSN PTBXL-Super | CPSC2018 |
|---|---|---|---|---|---|---|---|---|
| SimCLR | × | | 69.62 | 73.05 | 56.65 | 66.36 | 59.67 | 62.11 |
| BYOL | × | | 70.27 | 74.01 | 57.32 | 67.54 | 60.39 | 63.24 |
| BarlowTwins | × | | 68.98 | 72.85 | 55.97 | 65.89 | 58.76 | 61.35 |
| MoCo-v3 | × | | 69.41 | 73.29 | 56.54 | 66.12 | 59.82 | 62.07 |
| SimSiam | × | | 70.06 | 73.92 | 57.21 | 67.48 | 60.23 | 63.09 |
| TS-TCC | × | 100% | 71.32 | 75.16 | 58.47 | 68.34 | 61.55 | 64.48 |
| CLOCS | × | | 68.79 | 72.64 | 55.86 | 65.73 | 58.69 | 61.27 |
| ASTCL | × | | 69.23 | 73.18 | 56.61 | 66.27 | 59.74 | 62.12 |
| CRT | × | | 70.15 | 74.08 | 57.39 | 67.62 | 60.48 | 63.33 |
| STMEM | × | | 76.12 | **84.50** | 62.27 | 75.19 | 73.05 | 64.66 |
| MERL | √ | 0% | 88.21 | 78.01 | 76.77 | 76.56 | 74.15 | 82.86 |
| Ours | √ | 0% | **88.78** | 78.83 | **79.50** | **81.02** | **76.70** | **85.84** |

The target domain preparation is the same as the MERL (Liu et al. (2024b)), which is illustrated with details in the Appendix D. The experimental results are shown in Table 2. Notably, our DERI performs better across six distribution shift settings than the zero-shot method MERL, especially when the source domain is CPSC2018. Compared to self-supervised learning methods, only STMEM outperforms our DERI in the *PTBXL-Super to CSN* setting while DERI outperforms other methods with 100% training data for linear probing. It can be observed that MERL achieves the second-best performance on most settings, excluding *PTBXL-Super to CSN*. These experimental results support that multi-modal learning with ECG and reports for zero-shot classification can effectively improve the robustness and generalization of the learned representation. The significant performance advantages of DERI over MERL further illustrate the effectiveness of deepening modal interactions and enabling cross-modal representation learning by deepening the modal interactions of ECG-Report.

## 4.4 RESULTS ON REPORT GENERATION

In addition to the ability to learn valid representations of ECGs for cardiac condition classification, our proposed DERI method achieves deep cross-modal interactions, and the extracted representations contain valid clinical report information that can be used for diagnostic report generation. To further support our approach, we conducted experiments of report generation on the MIMIC-ECG dataset. The pre-trained DERI and MERL are used as the encoder to combine with DistilGPT2 as an encoder-decoder framework for report generation. To better verify the effect of the proposed mix encoding in DERI, we also adopt the aligned ECG encoding in DERI as a variant called DERI-align. The NLG and CE metrics of the generated reports are shown in Table 3.

Table 3: Report generation on MIMIC-ECG. Best results in bold and grey for second highest.

| Encoder | NLG | | | | | CE | | |
|---|---|---|---|---|---|---|---|---|
| | BLEU-1 | BLEU-2 | BLEU-3 | BLEU-4 | ROUGE-L | F1 | PRE | REC |
| MERL | 59.68 | 54.09 | 49.58 | 46.52 | 69.52 | 23.03 | 25.01 | 23.05 |
| DERI | **62.48** | **57.23** | **52.90** | **49.85** | **71.45** | **24.90** | **26.74** | **24.88** |
| DERI-Align | 61.33 | 55.98 | 51.75 | 48.50 | 70.59 | 23.33 | 24.96 | 23.60 |

We observe that our proposed DERI outperforms MERL on both NLG metrics and CE metrics, which illustrates that the representation learned by our method contains more clinical information. This result demonstrates that our proposed method enables better modal interactions between ECG

signals and REPORT, thus extracting more effective representations. In addition, by comparing the results of DERI-align with DERI, we demonstrate that the proposed cross-modal feature reconstruction method can effectively learn the information of the clinical report, thus generating a diagnostic report that is closer to the ground truth.

To better verify the CE metrics calculation method, we adopt large language models (LLMs) LLaMA2-7b (Touvron et al. (2023)) and vicuna-7b (Zheng et al. (2023)) to conduct report classification. Specifically, we feed the original reports and generated reports to the LLMs respectively and then ask the LLMs to choose the best class from six given categories, which include *Normal ECG, Myocardial Infarction, ST/T Change, Conduction Disturbance, Hypertrophy, and Others*. The answers of the original reports are regarded as ground truth and the answers of the generated reports are predicted labels. We then calculate the CE metrics as Figure 4.

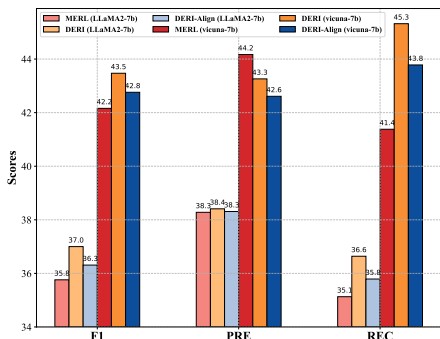

Figure 4: Report Generation CE Metrics.

We can observe that the results of report categorization using LLMs are basically the same as the results of our proposed zero-shot categorization method: DERI is the best and DERI-align is the second best, while both methods outperform MERL on F1. Meanwhile, the calculation of CE using vicuna performs better results than LLaMA2. We also provide example generated reports as Figure 5.

As we can see, The red portion of the ground truth is not accurately generated, the gray portion of the generated report is incorrect, and the green portion is correct. Although our method also does not fully generate the corresponding diagnostic reports, our method greatly reduces the error rate of the generated reports. From the results, the reports generated by our method are of higher quality, which also proves that our method achieves effective deep-modal interaction.

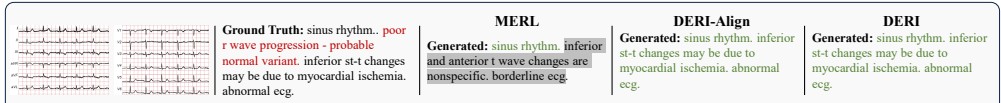

Figure 5: Example of generated reports on MIMIC-ECG dataset.

To conduct more comprehensive experiments about report generation, we adopt different language models as the text decoder instead of DistilGPT2, including Med-CPT (Jin et al. (2023)), PubMed-Bert (Gu et al. (2020)), and SciBert (Beltagy et al. (2019)). More details and experimental results can be found in Appendix E.

## 4.5 ABLATION STUDY

To better verify the performance of the key components/design choices of our DERI, we conduct comprehensive ablation studies on zero-shot classification and linear probing with 1% training data across different classification datasets. All results are proposed with **average AUC on six downstream datasets**.

Table 4: Ablating Multiple Alignment and Feature Reconstruction

| $\mathcal{L}_{align}$ | $\mathcal{L}_{mixed}$ | $\mathcal{L}_{DEC}^{e}$ | $\mathcal{L}_{DEC}^{t}$ | $\mathcal{L}_{RME}^{e}$ | Zero-shot | Linear Probing (1%) |
|---|---|---|---|---|---|---|
| √ | | √ | √ | √ | 76.34 | 72.36 |
| √ | √ | | √ | √ | 77.41 | 69.20 |
| √ | √ | √ | | √ | 78.06 | 69.61 |
| √ | √ | √ | √ | | 78.22 | 68.31 |
| √ | √ | √ | √ | √ | **78.73** | **74.05** |

**Multiple Alignment and Feature Reconstruction.** We realize the validation of the effect of these compositions by ablating the corresponding loss functions separately, and the experimental results are reported in Table 4. Table 4 shows that for zero-shot classification, $\mathcal{L}_{mixed}$ brings the best

improvement while $\mathcal{L}_{RME}^e$ for linear probing. This suggests that the second alignment of mixed encoding can better fuse the clinical information from reports because zero-shot classification learning is conducted to calculate the similarity between the learned representations and prompt representations. The effect of $\mathcal{L}_{RME}^e$ on linear probing exemplifies the effectiveness of the RME-Module we designed to improve the learning ability of ECG representations. The model trained with all the loss functions obtains the best performance, which illustrates the effectiveness of our proposed DERI method in ECG representation learning.

**RME-Module.** We compared the effect of the RME-Module and its variant that uses linear projectors instead of the attention mechanism which is set as RME-Linear, and Latent Dropout used by MERL. The experimental results are reported in Table 5. We can observe that the random masking strategy obtains better performance

Table 5: Ablating RME-Module.

|  | Zero-shot | Linear Probing (1%) |
| --- | --- | --- |
| Latent Dropout | 76.88 | 71.86 |
| RME-Linear | 78.05 | 72.40 |
| RME-Module | **78.73** | **74.05** |

than dropout while using the multi-head attention mechanism instead of global meaning can achieve the best performance, enhancing the model's ability to learn ECG representation for classification.

We then further explore the impact of the masking ratio $p$ on the performance by changing it from 0.1 to 0.5 with a step of 0.1. The experimental results are shown in Table 6. We observe that the masking ratio of 0.1 obtains the best performance among other masking ratios in both zero-shot classification and linear probing. Therefore, we adopt the masking ratio of 0.1 in our RME-Module.

Table 6: Ablating Masking Ratio.

| Mask-ratio | Zero-shot | Linear Probing (1%) |
| --- | --- | --- |
| 0.1 | **78.73** | 74.05 |
| 0.2 | 77.67 | 68.69 |
| 0.3 | 78.05 | 71.26 |
| 0.4 | 77.62 | 73.55 |
| 0.5 | 76.35 | 72.21 |

**Shared Embedding and Mix Encoding.** We also conduct experiments to verify the effect of the shared embedding used in cross-modal reconstruction. We remove the shared embedding from $\mathcal{D}t$ to $\mathcal{D}e$ as a variant of our proposed DERI. In addition, we use the pre-trained aligned ECG features to conduct zero-shot classification and linear probing to verify whether the mix encoding performs better than the aligned encoding for classification. This means that we adopt the same pre-training model but use the aligned ECG encoding instead of the mixed encoding for downstream tasks.

The experimental results are shown in Table 7. It can be observed that removing the shared embedding from $\mathcal{D}t$ to $\mathcal{D}e$ during text encoding reconstruction leads to a decline in model performance for both zero-shot and linear probing tasks. Furthermore, using the mixed encoding, which includes the decoded report features, outperforms using only the aligned ECG features. These findings underscore the strong representation learning capability of DERI.

Table 7: Shared Embedding and Mix Encoding.

|  | Zero-shot | Linear Probing (1%) |
| --- | --- | --- |
| Without SE | 77.25 | 70.22 |
| DERL-Align | 78.03 | 73.59 |
| DERL | **78.73** | **74.05** |

## 5 CONCLUSION

In this study, we proposed DERI, an innovative deep ECG-Report interaction framework for cross-modal representation learning. To obtain deep ECG-Report interaction, we design multiple alignments and cross-modal mutual reconstruction. Besides, an RME-Module is conducted on the ECG latent encoding for representation learning enhancement. Moreover, we extended ECG representation learning to clinical diagnostic report generation, aiming to deliver more intuitive ECG clinical insights. Our extensive experiments demonstrate the DERI's capability to learn the clinical semantics of ECG signals with the help of reports, which achieves the best performance on ECG classification and report generation.

**Limitations and Future Work.** One potential limitation of our work is that the report used is closer to a clinical semantic description of the signal and remains structurally different from a real diagnostic report. Additionally, we plan to expand our DERI into a more comprehensive cross-modal representation learning model, which can learn from other modal data, such as electronic medical records, further enhancing its relevance in clinical medicine.

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

# APPENDIX

## A    RELATED WORK ON ECG REPORT LEARNING

Exploration of ECG signals with the report is now more and more popular. Ref. (Yu et al. (2023)) is proposed to conduct ECG zero-shot classification based on LLMs and retrieval-augmented generation (RAG) and then obtain a great performance on ECG classification. However, this method needs to construct a vector database first for retrieval and the performance depends on the quality of the database. This RAG method has high storage requirements and high computing costs. In addition, although this method is also a multimodal method, it does not use the diagnostic report corresponding to the ECG signal, and cannot extract the high-level semantics in the report as efficiently as our DERI method.

METS (Li et al. (2024)) is proposed to simply calculate the cosine similarity between the ECG embedding and the report embedding for multi-modal ECG-Text SSL pre-training. However, this is too simple to learn the cross-modal representation of ECG and reports. Actually, it's the previous works of the MERL with simpler methods with lower results. MERL is the work that further improved METS.

MEIT (Wan et al. (2024)) aims to use an instruction prompt to generate reports based on the ECG signal input with LLMs. However, in our work, our DERI framework is not designed only for report generation, but also for ECG classification. It is used to learn effective representation with more clinical cardiac information from ECG signals. Although we also conduct report generation, it is just one of the ways to verify the representation of ECG learned.

ECG-Chat (Zhao et al. (2024)) is a great work for multi-modal ECG learning, which combines the ECG encoder and classification results to construct instructions for LLM. In addition to the ECG

signal, it also uses the electrical health record and other information for DSPy and GraphRAG. Compared to the original report, ECG-Chat can generate structured reports that include medical history diagnoses and recommendations.

## B  IMPLEMENTATION FOR ECG CLASSIFICATION.

We provide more details about ECG classification datasets used in our experiments here. We adopt the same data split strategy as MERL (Liu et al. (2024b)). The details are shown in Table 8. Notably, the PTBXL dataset with different tasks is split according to the official strategy (Wagner et al. (2020)) while the CPSC2018 and CSN datasets are split as 7:1:2.

Table 8: More details about the ECG classification datasets.

| Dataset | Number of Categories | Samples | Train | Valid | Test |
|---|---|---|---|---|---|
| PTBXL-Super | 5 | 21,388 | 17,084 | 2,146 | 2,158 |
| PTBXL-Sub | 23 | 21,388 | 17,084 | 2,146 | 2,158 |
| PTBXL-Form | 19 | 8,978 | 7,197 | 901 | 880 |
| PTBXL-Rhythm | 12 | 21,030 | 16,832 | 2,100 | 2,098 |
| CPSC2018 | 9 | 6,877 | 4,950 | 551 | 1,376 |
| CSN | 38 | 23,026 | 16,546 | 1,860 | 4,620 |

Furthermore, we provided more details about the hyper-parameters used for linear probing on ECG classification tasks in Table 9.

Table 9: Hyperparameter settings for Linear Probing

| | |
|---|---|
| Learning rate | 0.01 |
| Batch size | 16 |
| Epochs | 100 |
| Optimizer | AdamW |
| Learning rate scheduler | Cosine annealing |
| Warump steps | 5 |

## C  BASELINE.

More details about the compared baselines in our experiment are shown as below.

**SimCLR.** (Chen et al. (2020a)) This method aims to maximize consistency between differently augmented views of the same data example including random cropping, random distortion, and random Gaussian blur. Contrastive loss obtained by the representation of augmented views is used to optimize the base encoder.

**BYOL.** (Grill et al. (2020)) This method uses an online network to predict the presentation of other augmented views of the same sample obtained by a target network. Therefore, BYOL conducts contrastive learning without negative pairs.

**BarlowTwins.** (Zbontar et al. (2021)) This proposes to measure the cross-correlation matrix between the outputs obtained by two identical networks with distorted versions of a sample. The cross-correlation matrix is made as close to the identity matrix as possible since it can make the representation of distorted versions from the same simple to be similar.

**MoCo-v3.** (Chen et al. (2021)) This method proposes to train the transformer for self-supervised learning based on investigation of the fundamental components during training. In this way, they aim to overcome the instability of the transformer for a better representation.

**SimSiam.** (Chen & He (2021)) This method uses an encoder to process two augmented views of one sample and then a prediction MLP is applied on one side while stop-gradient is applied on the other side. Neither negative pairs nor momentum are used to learn meaningful representations.

**TS-TCC.** (Eldele et al. (2021)) This method first augments the sample using weak and strong augmentation and then learns robust temporal representation with a cross-view prediction task. Finally, the similarity among contexts from the same sample is minimized as contextual contrasting learning.

**CLOCS.** (Kiyasseh et al. (2021)) This method proposed to learn patient-specific representations of ECG signals via contrastive learning with consideration of both temporal and spatial information.

**ASTCL.** (Wang et al. (2023)) This method proposes a novel ECG augmentation method based on the noise attributes and then combines an adversarial module and a spatial-temporal contrastive module to learn the spatial-temporal and semantic representations of ECG signals.

**CRT.** (Zhang et al. (2023)) This method proposes to model temporal-spectral correlations of temporal time series by a cross-reconstruction transformer. Through cross-domain dropping reconstruction, the model can adequately capture the cross-domain correlations between temporal and spectral information of time series data.

**STMEM.** (Na et al. (2024)) This method is proposed to learn the spatial-temporal relations of ECG signals for masked modeling. ECG signals are divided into patches on the temporal and spatial dimensions and then the model reconstructs the masked patches to learn spatial-temporal features of 12-lead ECG signals for classification.

**MERL.** (Liu et al. (2024b)) This method extends ECG self-supervised learning to multi-modal learning by directly aligning the ECG signal encoding and the text encoding to learn ECG representation for classification. However, shallow interaction between ECG signals and reports can not provide effective clinical semantics for the representation.

## D  DISTRIBUTION SHIFT SETTING.

Considering that the source domain may not cover all the categories of the target domain, target domain categories are merged into the most similar source domain categories. Categories in the target domain while not in the source domain and the corresponding ECG recordings will be removed. The details of the category relations for distribution shift are illustrated in Table 10 and Table 11.

Table 10: PTBXL-Super to CPSC2018 and CSN.

| Source Domain | Target Domain | |
| --- | --- | --- |
| PTBXL-Super | CPSC2018 | CSN |
| HYP | None | RVH, LVH |
| NORM | NORM | SR |
| CD | 1AVB, CRBBB, CLBBB | 2AVB, 2AVB1, 1AVB, AVB, LBBB, RBBB, STDD |
| MI | None | MI |
| STTC | STE, STD | STTC, STE, TWO, STTU, QTIE, TWC |

Table 11: Category relations between CPSC2018 and CSN

| CPSC2018 | CSN |
| --- | --- |
| AFIB | AFIB |
| VPC | VPB |
| NORM | SR |
| 1AVB | 1AVB |
| CRBBB | RBBB |
| STE | STE |
| PAC | APB |
| CLBBB | LBBB |
| STD | STE, STTC, STTU, STDD |

## E  REPORT GENERATION.

We provided more comprehensive experimental results here by adopting different language models as the text decoder for report generation, including Med-CPT, PubMedBert, and SciBert. The ex-

perimental results of NLG and zero-shot CE are proposed in Table 12. We use the same strategy of R2Gen (Chen et al. (2020b)) to adopt these language models as text decoders for report generation.

Table 12: NLG and CE of Report Generation on MIMIC-ECG Dataset.

| Decoder | Encoder | NLG | | | | | CE | | |
|---|---|---|---|---|---|---|---|---|---|
| | | BLEU-1 | BLEU-2 | BLEU-3 | BLEU-4 | ROUGE-L | F1 | PRE | REC |
| DistilGPT2 | MERL | 59.68 | 54.09 | 49.58 | 46.52 | 69.52 | 23.03 | 25.01 | 23.05 |
| | DERI | **62.48** | **57.23** | **52.90** | **49.85** | **71.45** | **24.90** | **26.74** | **24.88** |
| | DERI-Align | 61.33 | 55.98 | 51.75 | 48.50 | 70.59 | 23.33 | 24.96 | 23.60 |
| PubMedBert | MERL | 57.87 | 51.84 | 46.99 | 43.44 | 71.58 | 28.31 | 31.39 | 28.98 |
| | DERI | **62.30** | **56.75** | **52.16** | **48.67** | **75.34** | **31.39** | **33.77** | **31.41** |
| | DERI-Align | 61.93 | 56.33 | 51.74 | 48.25 | 75.08 | 30.45 | 32.47 | 31.20 |
| MedCPT | MERL | 57.84 | 51.78 | 46.89 | 43.32 | 71.56 | 27.63 | 29.82 | 28.39 |
| | DERI | **61.11** | **55.42** | **50.79** | **47.28** | **74.36** | **31.27** | **33.73** | **31.24** |
| | DERI-Align | 61.88 | 56.28 | 51.68 | 48.19 | 75.03 | 28.33 | 30.44 | 29.76 |
| SciBert | MERL | 55.72 | 49.52 | 44.46 | 40.37 | 69.86 | 26.99 | 29.59 | 27.84 |
| | DERI | **61.95** | **56.37** | **51.78** | **48.33** | **75.11** | **31.17** | **33.98** | **30.99** |
| | DERI-Align | 61.50 | 55.84 | 51.17 | 47.58 | 74.95 | 28.57 | 30.14 | 29.58 |

We can observe that although DistilGPT2 achieves the best BLEU-n metrics, other decoders have better ROUGE-L and CE metrics, especially the PubMedBert. This may be attributed to PubMed-Bert being pre-trained on a dataset with biomedical research articles that contain more clinical information that is important to clinical reports. To better understand the zero-shot CE metrics, the whole process to obtain CE metrics here is provided in Figure 6.

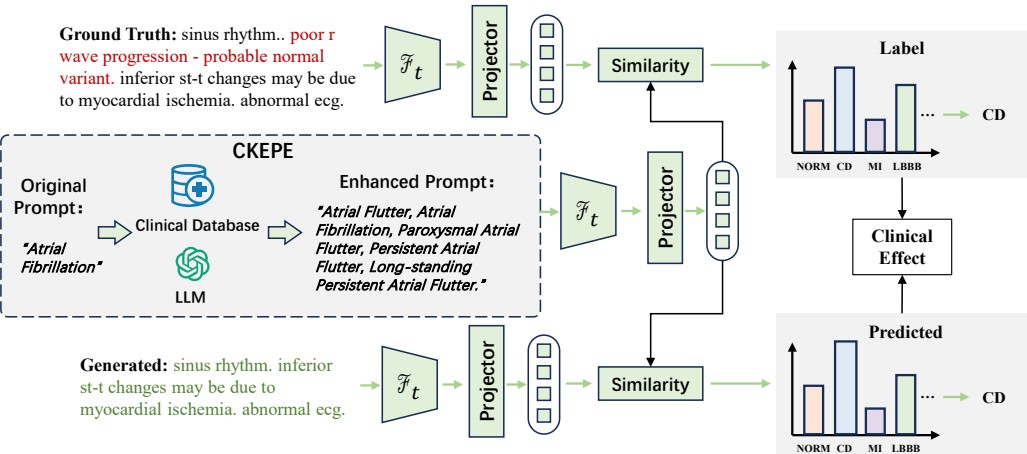

Figure 6: CE metrics calculation inspired by zero-shot classification.

We also provide more examples of the generated reports from MERL and DERI to make a better illustration, including reports in three lengths: long, medium, and short. The generated results are shown in Figure 7.

It can be observed that our DERI outperforms MERL in report generation at all three lengths, which indicates that our proposed DERI approach achieves a deeper ECG-Report cross-modal interaction, effectively incorporating the clinical semantics in the report while learning ECG representations.

To better verify the performance of our proposed DERI, we compared the result of the generated report task on the PTB-XL dataset. It should be noted that the results of baselines are referred to in the original paper and the experimental results are shown in Table 13.

We can see that our proposed DERI obtains the best performance, which verifies the great effect of our method on learning ECG-report cross-modal representation. In addition, LLM-based methods

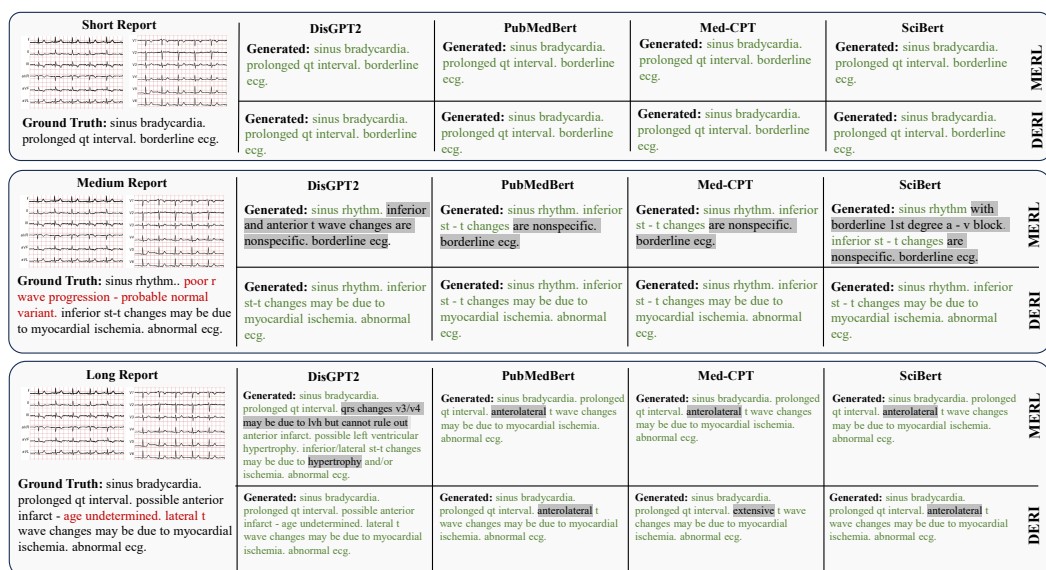

Figure 7: Examples of generated reports in three lengths.

Table 13: Report Generation Results on PTB-XL Dataset.

|  | Method | BLEU-1 | BLEU-2 | BLEU-3 | BLEU-4 | ROUGE-L |
|---|---|---|---|---|---|---|
| Our Framework | MERL-DisGPT2 | 48.9 | 44.4 | 41.2 | 37.4 | 55.4 |
|  | DERI-DisGPT2 | **58.6** | **54.8** | **51.9** | **48.6** | **64.2** |
|  | DERI-Align-DisGPT2 | 54.1 | 49.8 | 46.6 | 43.2 | 60.1 |
| Baseilnes from MEIT | GPT2-Medium | 32.9 | 27.8 | 25.4 | 23.2 | 39.1 |
|  | GPT2-Large | 43.7 | 39.5 | 35.5 | 32.0 | 48.1 |
|  | GPT-Neo | 47.4 | 44.9 | 39.8 | 37.3 | 48.6 |
|  | GPT-NeoX | 46.9 | 45.3 | 41.7 | 39.9 | 55.3 |
|  | GPT-J | 48.5 | 45.2 | 42.8 | 40.5 | 55.0 |
|  | BLOOM | 49.1 | 46.2 | 42.7 | 41.5 | 58.0 |
|  | OPT | 50.2 | 47.7 | 43.1 | 41.8 | 56.8 |
|  | LLaMA-1 | 51.4 | **48.5** | 46.5 | 43.0 | 58.8 |
|  | Mistral | 48.6 | 47.5 | 44.6 | 42.1 | 59.1 |
|  | LLaMA-2† | **51.5** | 48.4 | **46.9** | **43.9** | **59.4** |
|  | Mistral-Instruct† | 50.1 | 48.1 | 45.7 | 42.5 | 59.2 |
| Baseilnes from ECG-Chat | PTB-XL | 6.5 | - | - | 0.9 | 25.6 |
|  | ECG-Chat | 15.9 | - | - | 2.3 | 23.9 |
|  | ECG-Chat-DDP | **32.3** | - | - | **11.2** | **29.9** |

always need high computing resources, while our proposed method can be conducted on 4090 GPUs with greater performance.

Moreover, we conduct report classification on PTBXL for better support that our method can effectively learn high-level semantics from report data, including single-label and multi-label. Specifically, we conduct single-label classification as the same as Figure 6. We use the Med-CPT to encode the prompt of all categories as the label embedding first. Then, to obtain ground truth, we encode the original report to obtain embedding and then calculate the similarity of all label embedding. The category with the highest similarity will be regarded as the true label of the original report. For the predicted label, we conduct the same process with the generated report. On the other hand, for multi-label report classification, since the real label of ECG signals in the downstream dataset is provided, we feed the corresponding prompt of the target category of the label to the text encoder to obtain the prompt embedding. Then, after our DERI learning the representation of the ECG signals, we calculate the similarity between the learned representation and the target prompt embedding and

then conduct a sigmoid function on the similarity to obtain the predicted probability of each target category. Finally, we conduct an optimal classification threshold search with the help of a precision-recall curve. Categories with a higher probability than the threshold will be regarded as predicted labels. Experimental results are shown in Table 14.

Table 14: Report Classification on PTBXL dataset.

| Single Label | Method | F1 | PRE | REC |
|---|---|---|---|---|
| Prompt | MERL-DisGPT2 | 22.6 | 27.9 | 21.5 |
| | DERI-DisGPT2 | **40.1** | **46.5** | **38.7** |
| | DERI-Align-DisGPT2 | 32.0 | 38.1 | 30.0 |

| **Multi-label** | **Method** | **F1** | **Acc** | **AUC** |
|---|---|---|---|---|
| Super | MERL-DisGPT2 | 53.3 | 66.0 | 75.7 |
| | DERI-DisGPT2 | **56.1** | **72.9** | **76.9** |
| | DERI-Align-DisGPT2 | 55.5 | 72.3 | 76.1 |
| Sub | MERL-DisGPT2 | 19.3 | 85.0 | 71.1 |
| | DERI-DisGPT2 | **21.1** | **86.5** | **72.7** |
| | DERI-Align-DisGPT2 | 19.7 | 85.3 | 72.2 |
| Form | MERL-DisGPT2 | 20.8 | 78.3 | 62.9 |
| | DERI-DisGPT2 | **26.5** | **89.0** | **68.0** |
| | DERI-Align-DisGPT2 | 24.8 | 84.0 | 66.0 |
| Rhythm | MERL-DisGPT2 | 18.5 | 93.3 | 71.1 |
| | DERI-DisGPT2 | **24.1** | **95.2** | **74.5** |
| | DERI-Align-DisGPT2 | 23.1 | 94.0 | 73.7 |

## F  REPRESENTATION VISUALIZATION.

To investigate the learned ECG representation further, we visualize the learned representation of DERI and MERL. We use t-SNE to visualize the representation as Figure 8. For PTBXL, we adopt the PTBXL-Super setting to obtain the category. For Chapman, we keep the nine categories with the largest sample sizes for better visualization. It can be observed that after t-SNE, the representations learned by our DERI can be more clustered, with greater differentiation between different categories, which indicates that our method is able to learn discriminative ECG representations more efficiently than the simple utilization of diagnostic reports by MERL, containing more cardiac clinical information.

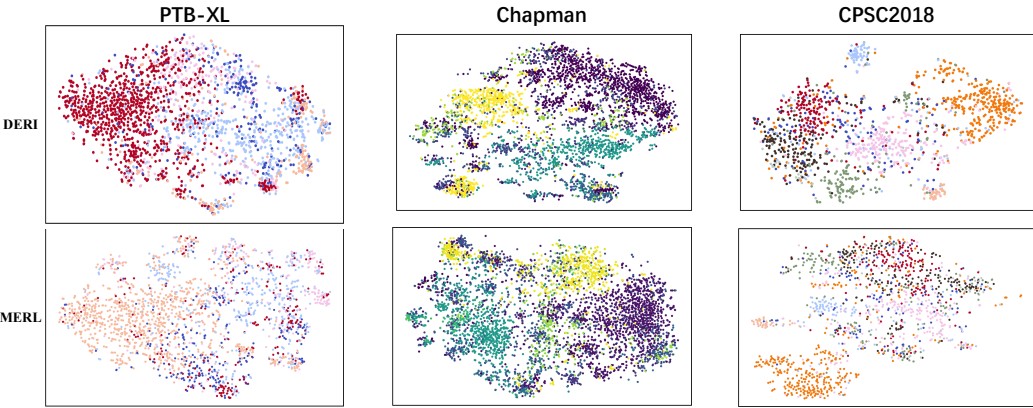

Figure 8: T-SNE visualization on three classification datasets.

