# OpenReview forum: "Deep ECG-Report Interaction Framework for Cross-Modal Representation Learning"
_ICLR.cc/2025/Conference — ICLR 2025 Conference Withdrawn Submission_

### Official Review · Reviewer_5ABa · 2024-10-27

**Soundness:** 3
**Presentation:** 3
**Contribution:** 2
**Rating:** 3
**Confidence:** 5

**Summary:**

This paper proposes the Deep ECG-Report Interaction (DERI) framework to address the lack of clinical semantics in ECG representation learning.
By integrating ECG signals with clinical reports using multi-level alignment strategies, DERI enhances cross-modal learning. It also incorporates a language model for ECG report generation, demonstrating superior performance across multiple datasets.

**Strengths:**

Align multi-level features using contrastive loss, while also performing cross-modal reconstruction by using ECG signals to reconstruct text and text to reconstruct ECG. This approach enables learning mutual information between the two modalities. The framework is evaluated across multiple datasets and methods for comprehensive assessment. However, the compared baseline methods are limited, and the evaluation metrics are ambiguous.

**Weaknesses:**

- The novelty is limited. In Section 3.2, the work uses cross-modal alignment with the original contrastive loss. Section 3.3's approach for cross-modal reconstruction is similar to what was proposed in MRM [1], even though MRM was originally for the image domain. The method in this work closely resembles MRM. Moreover, MRM uses features extracted from masked inputs to reconstruct the other modality, while DERI uses features extracted directly from the original inputs, which reduces the difficulty of the reconstruction task.
- There is a lack of baseline and comparison in the report generation task. In MEIT [2], a comprehensive benchmark for ECG report generation is proposed and implemented on the MIMIC-ECG and PTB-XL datasets, both of which are also used in DERI. However, the authors do not compare DERI against any baseline from MEIT and only use GPT-2 as the text decoder, which is outdated, having been released in 2019.
- The reproducibility issue is further compounded by the authors' apparent reluctance to share their code.
- The report generation task is not implemented on PTB-XL. Since MIMIC-ECG is used for pretraining, evaluating solely on MIMIC-ECG does not sufficiently assess generalizability and robustness, as all the data is seen during pretraining.
- The evaluation metric for ECG report generation is lacking. The evaluation metric for clinical efficacy is ambiguous.

[1] Zhou, Hong-Yu, et al. "Advancing Radiograph Representation Learning with Masked Record Modeling." The Eleventh International Conference on Learning Representations.
[2] Wan, Zhongwei, et al. "MEIT: Multi-Modal Electrocardiogram Instruction Tuning on Large Language Models for Report Generation." arXiv preprint arXiv:2403.04945 (2024).

**Questions:**

- In Appendix Section D, Table 12, the authors include various text models for report generation, including encoder-only models like MedCPT (desgined for text retrieval task). Using encoder-only models for report generation is questionable, as it conflicts with the mainstream approach seen in works such as MEIT [1] and RGRG [2], which typically utilize encoder-decoder or decoder-only models for text generation tasks. Encoder-only models are not designed for generative tasks like report generation, so their inclusion deviates from standard practices.

- Regarding the computation of clinical efficacy in Section 4.2, several aspects need clarification:
(1) **How is the prompt embedding obtained from the decoder?** If the decoder is used to obtain prompt embeddings, is it based on the representation from the [EOS] token? Clarification is needed on how the embedding is extracted from a decoder-only architecture.
(2) **How is the classification probability for categories computed using a text decoder?** Does this refer to the highest probability assigned to the category name token? Some diseases (e.g., "myocardial infarctions") are tokenized into multiple tokens. If this is the case, how is the classification probability determined for multi-token categories?
(3) **Handling multiple classes in ECGs**: The PTB-XL dataset shows that ECGs can belong to multiple classes simultaneously. If the authors use only the highest probability for classification, they may be reducing the prediction to a single class, which ignores other relevant conditions. Why are additional classes not considered in the evaluation?
(4) **Discrepancy between prompts and generated reports**: The method uses a single-class description as the prompt for classification, whereas the generated report may describe multiple conditions associated with the ECG signal. There is a clear gap between the prompt (single class) and the report (multi-class). How is this gap addressed in the evaluation process?

[1] Wan, Zhongwei, et al. "MEIT: Multi-Modal Electrocardiogram Instruction Tuning on Large Language Models for Report Generation." arXiv preprint arXiv:2403.04945 (2024). Tanida, Tim, et al. "Interactive and explainable region-guided radiology report generation." Proceedings of the IEEE/CVF Conference on Computer Vision and Pattern Recognition. 2023.

---

> ### Author Response · Authors · 2024-11-17
> **Response to Reviewer 5ABa (1)**
>
> **Q4.1**:  About the novelty and discussion with MRM.
>
> **A4.1**: We highlight our main innovations in A3.3 (rebuttal to Reviewer 3). In addition, MEM is proposed to learn knowledge-enhanced semantic representations of radiograph with a multi-task scheme including radiograph self-completion and report completion. Although MEM introduces image features into the task of report completion, it is still essentially a report mask generation task. The role of image features is to inject some visual features to assist mask prediction of reports. This task can not be regarded as a cross-modal reconstruction task because it mainly completes the report by covering the text representation. Our DERI architecture aims to reconstruct another modal representation based on ECG representation and report representation, respectively. This process of cross-modal mutual reconstruction in our DERI is designed to guide the model towards deeper modal interactions, allowing the ECG representation to effectively learn high-level semantics in the report. In terms of motivation and implementation method design, our approach is very different from MEM, not similar. On the other hand, MEM is a mask reconstruction task, so the original representation of the modes is obscured. However, the refactoring of MEM is actually done using the representation of the same mode, so this does not mean that such a refactoring process is more difficult than our method. Our method is to achieve cross-modal reconstruction, and the model needs to overcome the direct distribution gap of different modes, which is often more difficult than the reconstruction of the original mode. To solve this problem, we put cross-modal refactoring behind the cross-modal alignment phase to achieve better refactoring. This effective combination of multimodal alignment and reconstruction helps us dig deeper into the relationship between ECG and report to learn effective representations with more clinical semantics.
>
> **Q4.2**: A lack of baseline and comparison in the report generation task, comparison with MEIT, and discussion about GPT-2.
>
> **A4.2**: We conducted more experiments in the report generation tasks, including comparison with MEIT, and the results are shown in the Table below.
>
> |               |     Method                | BLEU-1 | BLEU-2 | BLEU-3 | BLEU-4 | ROUGE-L  |
> |---------------|--------------------|--------|--------|--------|--------|----------|
> | Our Framework | MERL-DisGPT2       | 48.9   | 44.4   | 41.2   | 37.4   | 55.4     |
> |               | DERI-DisGPT2       | **58.6**  |**54.8**   | **51.9**   |**48.6**   | **64.2**     |
> |               | DERI-Align-DisGPT2 | 54.1   | 49.8   | 46.6   | 43.2   | 60.1     |
> |---------------|--------------------|--------|--------|--------|--------|----------|
> | Ref [A]       | GPT2-Medium        | 32.9   | 27.8   | 25.4   | 23.2   | 39.1     |
> |               | GPT2-Large         | 43.7   | 39.5   | 35.5   | 32.0   | 48.1     |
> |               | GPT-Neo            | 47.4   | 44.9   | 39.8   | 37.3   | 48.6     |
> |               | GPT-NeoX           | 46.9   | 45.3   | 41.7   | 39.9   | 55.3     |
> |               | GPT-J              | 48.5   | 45.2   | 42.8   | 40.5   | 55.0     |
> |               | BLOOM              | 49.1   | 46.2   | 42.7   | 41.5   | 58.0     |
> |               | OPT                | 50.2   | 47.7   | 43.1   | 41.8   | 56.8     |
> |               | LLaMA-1            | 51.4   | 48.5   | 46.5   | 43.0   | 58.8     |
> |               | Mistral            | 48.6   | 47.5   | 44.6   | 42.1   | 59.1     |
> |               | LLaMA-2†           | 51.5   | 48.4   | 46.9   | 43.9   | 59.4     |
> |               | Mistral-Instruct†  | 50.1   | 48.1   | 45.7   | 42.5   | 59.2     |
> |---------------|--------------------|--------|--------|--------|--------|----------|
> | Ref [B]       | PTB-XL             | 6.5    | -      | -      | 0.9    | 25.6     |
> |               | ECG-Chat           | 15.9   | -      | -      | 2.3    | 23.9     |
> |               | ECG-Chat-DDP       | 32.3   | -      | -      | 11.2   | 29.9     |
>
> Although GPT-2 was proposed in 2019, it is now still widely used in clinical report generation as we discussed in our paper. Ref. [A] (2023) and Ref. [B] (2024) have verified the great performance of GPT-2 on medical report generation. In addition, compared to LLM such as LLaMA, GPT-2 needs much fewer computing resources. And our experimental results on PTB-XL compared with new baselines with LLMs showed that our method with GPT-2 obtained the best performance. Therefore, using GPT-2 as the text decoder is not outdated.

---

> > ### Author Response · Authors · 2024-11-17
> > **Response to Reviewer 5ABa (2)**
> >
> > **Reference used in A4.2**:
> >
> > [A] Wan, Zhongwei, et al. "Electrocardiogram instruction tuning for report generation." arXiv preprint arXiv:2403.04945 (2024).
> >
> > [B] Zhao, Yubao, et al. "ECG-Chat: A Large ECG-Language Model for Cardiac Disease Diagnosis." arXiv preprint arXiv:2408.08849 (2024).
> >
> > [C] Yuan Liu, Songyang Zhang, Jiacheng Chen, Zhaohui Yu, Kai Chen, and Dahua Lin. Improving pixel-based mim by reducing wasted modeling capability. In Proceedings of the IEEE/CVF In- ternational Conference on Computer Vision, pp. 5361–5372, 2023b.
> >
> > [D] Wang, F., Du, S., & Yu, L. (2024). HERGen: Elevating Radiology Report Generation with Longitudinal Data. arXiv preprint arXiv:2407.15158.
> >
> > **Q4.3**: The reproducibility issue is further compounded by the authors' apparent reluctance to share their code.
> >
> > **A4.3**: This is a serious misunderstanding of our work, and in the supplement, we have submitted our core source code for reference. We will also make all of our code public on GitHub as soon as the article is accepted.
> >
> > **Q4.4**: The report generation task is not implemented on PTB-XL. Since MIMIC-ECG is used for pretraining, evaluating solely on MIMIC-ECG does not sufficiently assess generalizability and robustness, as all the data is seen during pretraining.
> >
> > **A4.4**: Thanks for your valuable comments. We have conducted more experiments on PTB-XL about report generation and the results are provided in the table before (refer to A4.2). Our method still obtains the best performance than baselines, showing the great generalizability and robustness of our method.
> >
> > **Q4.5**: The evaluation metric for ECG report generation.
> >
> > **A4.5**: We evaluate the generated ECG report from two aspects: Natural Language Generation (NLG) and d Clinical Efficiency (CE).
> >
> > For NLG, we adopt BLEU-n for n-gram overlap evaluation and ROUGE-L with the longest common sub-sequence between the original report and generated report.
> >
> > For CE, we adopt two ways to better evaluate the generated reports. The process of obtaining CE for x-ray report generation requires a pre-trained ChestXRayBERT to classify the original report and generated report as the true label and predicted label respectively. However, there is no pre-trained language model for ECG report classification. Therefore, on the one hand, we adopt open-source LLM including LLaMA2-7b and vicuna-7b as the classifier of the ECG report. The LLMs are used to classify the original report and generated report from six given categories, which include Normal ECG, Myocardial Infarction, ST/T Change, Conduction Disturbance, Hypertrophy, and Others. The category of the original report is regarded as the true label and the one of the generated report is the predicted label. We then use these labels to calculate the CE metrics, which are shown in Figure 4 in our paper.
> >
> > On the other hand, inspired by the zero-shot classification method of ECG signal, we conduct report zero-shot classification, as illustrated in Appendix D and Figure 6 in our paper. Specifically, we use the Med-CPT to encode the prompt of all categories as the label embedding first. Then, to obtain ground truth, we encode the original report to obtain embedding and then calculate the similarity of all label embedding. The category with the highest similarity will be regarded as the true label of the original report. For the predicted label, we conduct the same process with the generated report.
> >
> > Furthermore, for PTB-XL with annotated labels, we conduct more experiments to verify the CE of the proposed DERI framework. The results are shown below.
> >
> > | Single Label | Method             | F1    | PRE   | REC    |
> > |--------------|--------------------|-------|-------|--------|
> > | Prompt       | MERL-DisGPT2       | 22.6  | 27.9  | 21.5   |
> > |              | DERI-DisGPT2       | 40.1  | 46.5  | 38.7   |
> > |              | DERI-Align-DisGPT2 | 32.0  | 38.1  | 30.0   |
> > |--------------|--------------------|-------|-------|--------|
> > | Multi-label  | Method             | F1    | Acc   | AUC    |
> > | Super        | MERL-DisGPT2       | 53.3  | 66.0  | 75.7   |
> > |              | DERI-DisGPT2       | 56.1  | 72.9  | 76.9   |
> > |              | DERI-Align-DisGPT2 | 55.5  | 72.3  | 76.1   |
> > |--------------|--------------------|-------|-------|--------|
> > | Sub          | MERL-DisGPT2       | 19.3  | 85.0  | 71.1   |
> > |              | DERI-DisGPT2       | 21.1  | 86.5  | 72.7   |
> > |              | DERI-Align-DisGPT2 | 19.7  | 85.3  | 72.2   |
> > |--------------|--------------------|-------|-------|--------|
> > | Form         | MERL-DisGPT2       | 20.8  | 78.3  | 62.9   |
> > |              | DERI-DisGPT2       | 26.5  | 89.0  | 68.0   |
> > |              | DERI-Align-DisGPT2 | 24.8  | 84.0  | 66.0   |
> > |--------------|--------------------|-------|-------|--------|
> > | Rhythm       | MERL-DisGPT2       | 18.5  | 93.3  | 71.1   |
> > |              | DERI-DisGPT2       | 24.1  | 95.2  | 74.5   |
> > |              | DERI-Align-DisGPT2 | 23.1  | 94.0  | 73.7   |

---

> > > ### Author Response · Authors · 2024-11-17
> > > **Response to Reviewer 5ABa (3)**
> > >
> > > **Q4.6**: About Med-CPT encoder for report generation.
> > >
> > > **A4.6**: Actually, our method is designed with GPT-2 as the text decoder for report generation. We tried to adopt Med-CPT as one of the decoder baselines since it was trained with more biomedical information. The model has more medically relevant corpus information, which may be beneficial for understanding medical reports. Since it is an encoder-only model, we adopt the method of R2Gen (Ref. [A] in EMNLP 2020) and HERGen (Ref. [B]) to conduct report generation, which actually just inherits the Med-CPT corpus vocabulary and tokenizer, and then retrained the transformer model on our MIME-ECG dataset. The specific process can be referred to in our code in supplementary materials or the GitHub project of Ref.[A] and Ref.[B]. We don’t directly use the encoder-only model to conduct report-generation tasks.
> > >
> > > **Q4.7**: About the prompt embedding.
> > >
> > > **A4.7**: Given the prompt text of each category, we use the pre-trained **Med-CPT encoder** to obtain the prompt embedding, which is further fine-tuned during the pre-training of our DERI. Med-CPT is an encoder-only architecture, which generates embeddings of biomedical texts that can be used for semantic search (dense retrieval). It has been pre-trained on the biomedical corpus, which makes it suitable for our task.
> > >
> > > **Q4.8**: About the classification probability and predicted label.
> > >
> > > **A4.8**: As we clearly illustrated in our paper (lines 274-275 on page 6), we obtain the prompt embedding of each category and the embedding of the original report and generated report respectively with the trained **text encoder**, rather than using a text decoder. For CE of report generation, with the help of a text encoder, we can obtain the embedding and then calculate the similarity between embedding, and then decide the classification results, as in **A4.5**. For zero-shot classification, since the real label of ECG signals in the downstream dataset is provided, we feed the corresponding prompt of the target category of the label to the text encoder to obtain the prompt embedding. Then, after our DERI learning the representation of the ECG signals, we calculate the similarity between the learned representation and the target prompt embedding and then conduct a sigmoid function on the similarity to obtain the predicted probability of each target category. Finally, we conduct an optimal classification threshold search with the help of a precision-recall curve. Categories with a higher probability than the threshold will be regarded as predicted labels.
> > >
> > > **Q4.9**: Handling multiple classes in ECGs.
> > >
> > > **A4.9**: For multi-label ECG signals classification in PTB-XL, our zero-shot classification will not only consider the category with the highest probability as we illustrate in A4.5. For multi-label ECG classification, our DERI use the learned representation of ECG signals to calculate the similarity with prompt embedding of targeted categories. After conducting an optimal classification threshold search, all categories above this threshold are considered to be predicted, so our method can effectively solve the problem of multi-label ECG zero-shot classification.
> > >
> > > **Q4.10**: Discrepancy between prompts and generated reports.
> > >
> > > **A4.10**: Since there is no corresponding label of the reports in the MIMIC-ECG dataset, we simply regard report classification as a single-class classification, which is not similar to ECG zero-shot classification. On the one hand, without the help of a cardiologist, it is hard to label the reports with multiple conditions. In this case, we choose the category with the highest similarity as the label to be closer to the corresponding category of the report, thus improving the accuracy of the report classification. On the other hand, the classification of reports is not the core innovation of our approach, the CE calculation of the generated reports is to better evaluate whether our approach is as in-depth as we claim to be in the modal interactions between the ECG-reports, and thus learn more about the advanced clinical semantic information contained in the reports. The experimental results of our comparison with MERL also prove the effectiveness of our method.
> > > Moreover, to better evaluate whether our method can deal with multi-label reports, we conduct multi-label report classification experiments on the PTB-XL dataset. We regard the multiple labels of the ECG signals as the target labels and use them to obtain the targeted prompt embedding. Then we also calculate the similarity with prompt embedding of targeted categories and conduct optimal classification threshold search as we do for multi-label ECG zero-shot classification. The experimental results are shown in A4.5 before.
> > >
> > > We hope our response has addressed all concerns. We would greatly appreciate any further constructive comments or discussions.

---

### Official Review · Reviewer_pgv7 · 2024-11-02

**Soundness:** 2
**Presentation:** 2
**Contribution:** 1
**Rating:** 3
**Confidence:** 5

**Summary:**

The paper proposes the Deep ECG-Report Interaction (DERI) framework, a novel method for cross-modal representation learning that combines ECG with clinical reports. This paper introduces cross-modal alignment for representation learning and RME module for enhanced ECG learning.

**Strengths:**

The paper is well-organized and generally easy to follow. The flow from the motivation behind the DERI framework to the detailed explanation of its architecture, followed by experiments and results, is logical and well-structured. The diagrams, particularly those illustrating the DERI architecture and its training process, are helpful in understanding the complex cross-modal interactions.

The technical descriptions, such as the use of multiple alignments, the RME module, and the integration of language models for report generation, are well-explained.

**Weaknesses:**

The paper shows a lack of understanding of related work, with many previous related articles not cited or discussed. The following articles [1-4] need to be added and discussed in the paper.

Several methods from the referenced articles need to be used as baselines and compared in the experimental section, especially for the ECG report generation part.

There are many similarities between this paper and the article "Zero-Shot ECG Classification with Multimodal Learning and Test-time Clinical Knowledge Enhancement (MERL)," with a lack of innovation.

For instance, two of the losses used in the paper are almost identical to the ones used in MERL (CLIP loss and mask loss); the paper merely describes them in a different way. The report generation method is also quite similar to many multimodal approaches, such as the BLIP method, and does not represent true innovation. Moreover, these papers have not been cited.

The downstream task system is also very similar to MERL, except for report generation. However, many report generation baselines are missing from this paper.

[1] Wan, Zhongwei, et al. "Electrocardiogram instruction tuning for report generation." arXiv preprint arXiv:2403.04945 (2024).

[2] Li, Jun, et al. "Frozen language model helps ecg zero-shot learning." Medical Imaging with Deep Learning. PMLR, 2024.

[3] Yu, Han, Peikun Guo, and Akane Sano. "Zero-shot ECG diagnosis with large language models and retrieval-augmented generation." Machine Learning for Health (ML4H). PMLR, 2023.

[4] Zhao, Yubao, et al. "ECG-Chat: A Large ECG-Language Model for Cardiac Disease Diagnosis." arXiv preprint arXiv:2408.08849 (2024).

**Questions:**

Please supplement the references and baseline methods [1-4] in the experiments, and fully discuss and compare their advantages, disadvantages, and innovations in the paper.

Please explain the parts that are overly similar to MERL and highlight the points of technological innovation.

---

> ### Author Response · Authors · 2024-11-17
> **Response to Reviewer pgv7 (1)**
>
> We appreciate your review and the positive comments regarding our paper. We would like to respond to your comments as follows.
>
> **Q3.1**: The paper shows a lack of understanding of related work, with many previous related articles not cited or discussed. The following articles [1-4] need to be added and discussed in the paper (compare their advantages, disadvantages, and innovations in the paper).
>
> **A3.1**: Thanks for your valuable comment. We have discussed these related works below:
>
> **Ref [1]** aims to use an instruction prompt to generate reports based on the ECG signal input with LLMs. However, in our work, our DERI framework is not designed only for report generation, but also for ECG classification. It is used to learn effective representation with more clinical cardiac information from ECG signals. Although we also conduct report generation, it is just one of the ways to verify the representation of ECG learned. In addition, the experiments are conducted on 4090 GPUs with less computing resources. To better compare our method and the MEIT framework in Ref [1], we add report generations on the PTB-XL dataset, which are not used in our pre-training process. The compared results are shown in the following table below with Ref [4].
>
> **Ref [2]** proposes a method called METS to simply calculate the cosine similarity between the ECG embedding and the report embedding for multi-modal ECG-Text SSL pre-training. However, this is too simple to learn the cross-modal representation of ECG and reports. Actually, it’s the previous works of the MERL with simpler methods with lower results. MERL is the work further improved by the authors of Ref [2], so we did not focus on this paper when we selected the baseline method.
>
> **Ref [3]** is proposed to conduct ECG zero-shot classification based on LLMs and RAG and then obtain a great performance on ECG classification. However, this method needs to construct a vector database first for retrieval and the performance depends on the quality of the database. This RAG method has high storage requirements and high computing costs. In addition, although this method is also a multimodal method, it does not use the diagnostic report corresponding to the ECG signal, and cannot extract the high-level semantics in the report as efficiently as our DERI method. However, the use of some baseline feature engineering for ECG signals in this method also gives us inspiration for further work. To better compare our method with this method, we compared the classification performance on PTB-XL, as shown in the table below:
> |              | Supervised | Few-shot TNP |    Few-shot TNP        | Zero-shot RAG |         Zero-shot RAG   |    Zero-shot RAG     | Linear Probing DERI |     Linear Probing DERI   |    Linear Probing DERI    | Zero-shot  |    Zero-shot     |
> |--------------|------------|--------------|------------|---------------|------------|---------|---------------------|-------|-------|------------|--------|
> | PTB-XL-Super | 1D-CNN     | LLaMA2-7B    | LLaMA2-13B | LLaMA2-7B     | LLaMA2-13B | GPT-3.5 | 1%                  | 10%   | 100%  | DERI       | MERL   |
> | Macro F1     | 66.0       | 35.7         | 34.8       | 61.7          | 62.2       | 66.9    | 65.6                | 70.9  | 72.4  | 55.4       | 53.3   |
> | Accuracy     | 74.8       | 41.7         | 42.2       | 71.4          | 72.6       | 75.7    | 84.3                | 87.2  | 87.6  | 74.9       | 71.9   |
>
> **Ref [4]** proposes ECG-Chat, which is a great work for multi-modal ECG learning. We didn’t discuss it since it was released on Arxiv less than 2 months before our work was submitted to ICLR. It combines the ECG encoder and classification results to construct instructions for LLM. In addition to the ECG signal, it also uses the electrical health record and other information for DSPy and GraphRAG. Compared to the original report, ECG-Chat can generate structured reports that include medical history diagnoses and recommendations. Therefore, to compare our DERI with ECG-Chat, we compared the result of the generated report task on the PTB-XL dataset with ECG-Chat. The additional experimental results with these baselines on ECG report generation. It should be noted that the results of baselines are referred to in the original paper.

---

> > ### Author Response · Authors · 2024-11-17
> > **Response to Reviewer pgv7 (2)**
> >
> > **Table following Ref.[4] in A3.1**
> > |               |     Method                | BLEU-1 | BLEU-2 | BLEU-3 | BLEU-4 | ROUGE-L  |
> > |---------------|--------------------|--------|--------|--------|--------|----------|
> > | Our Framework | MERL-DisGPT2       | 48.9   | 44.4   | 41.2   | 37.4   | 55.4     |
> > |               | DERI-DisGPT2       | **58.6**  |**54.8**   | **51.9**   |**48.6**   | **64.2**     |
> > |               | DERI-Align-DisGPT2 | 54.1   | 49.8   | 46.6   | 43.2   | 60.1     |
> > |---------------|--------------------|--------|--------|--------|--------|----------|
> > | Ref [1]       | GPT2-Medium        | 32.9   | 27.8   | 25.4   | 23.2   | 39.1     |
> > |               | GPT2-Large         | 43.7   | 39.5   | 35.5   | 32.0   | 48.1     |
> > |               | GPT-Neo            | 47.4   | 44.9   | 39.8   | 37.3   | 48.6     |
> > |               | GPT-NeoX           | 46.9   | 45.3   | 41.7   | 39.9   | 55.3     |
> > |               | GPT-J              | 48.5   | 45.2   | 42.8   | 40.5   | 55.0     |
> > |               | BLOOM              | 49.1   | 46.2   | 42.7   | 41.5   | 58.0     |
> > |               | OPT                | 50.2   | 47.7   | 43.1   | 41.8   | 56.8     |
> > |               | LLaMA-1            | 51.4   | 48.5   | 46.5   | 43.0   | 58.8     |
> > |               | Mistral            | 48.6   | 47.5   | 44.6   | 42.1   | 59.1     |
> > |               | LLaMA-2†           | 51.5   | 48.4   | 46.9   | 43.9   | 59.4     |
> > |               | Mistral-Instruct†  | 50.1   | 48.1   | 45.7   | 42.5   | 59.2     |
> > |---------------|--------------------|--------|--------|--------|--------|----------|
> > | Ref [4]       | PTB-XL             | 6.5    | -      | -      | 0.9    | 25.6     |
> > |               | ECG-Chat           | 15.9   | -      | -      | 2.3    | 23.9     |
> > |               | ECG-Chat-DDP       | 32.3   | -      | -      | 11.2   | 29.9     |
> >
> > **Q3.2**: Several methods from the referenced articles need to be used as baselines and compared in the experimental section, especially for the ECG report generation part.
> >
> > **A3.2**: We have provided more baselines for the ECG report generation part as the table in A3.1. It should be noted that we compared the results on the PTB-XL dataset since the pre-training process of our DERI is conducted without the PTB-XL dataset.
> >
> > **Q3.3**: There are many similarities between this paper and the article "Zero-Shot ECG Classification with Multimodal Learning and Test-time Clinical Knowledge Enhancement (MERL)," with a lack of innovation.
> >
> > **A3.3**: MERL conducts simple Cross-Modal Alignment between ECG and report in the feature space simply to learn representation. However, the interaction between modalities is relatively shallow, which cannot effectively convey the high-level semantics of ECG recordings. Simple alignment only makes the learned ECG representation close to the reported representation in the potential space, but it does not effectively learn the cardiac information contained in the reported representation. Compared to it, our proposed method aims to explore deep modality interaction between ECG signals and clinical reports with the combination of contrastive learning (alignment) and generative learning (reconstruction), which is a much different approach than MERL. We designed our framework to enable the model to learn more effective representation, which contains more high-level clinical semantics from reports, of ECG signals. The innovation of our work can be summarized as follows:
> >
> > a. To learn effective ECG representation for cardiac conditions from reports, we propose a novel cross-modal framework of ECG-Report via multiple feature alignment and mutual feature reconstruction. The combination of alignment and reconstruction in different levels enables the model to conduct deep interaction between ECG and report, which enables the learned representations to contain more effective high-level semantics, integrating features across modalities. This approach strengthens the representation's ability to capture the clinical context, enhancing both accuracy and relevance for diagnosis.
> >
> > b. A novel framework to combine the ECG encoder with the language model for ECG report generation is proposed in our method, in which the parameters of the encoder are frozen and just the language model is finetuned. On the one hand, our DERI can easily verify whether the learned representation contains high-level semantics from clinical reports. On the other hand, we used smaller models such as GPT2 to achieve report generation and achieved much more accurate results on the PTB-XL dataset than large models such as LLaMA and other methods like RAG (the result can be seen in the table in A3.1). This means that we have designed a report generation framework that requires only a small amount of computing resources to match or exceed LLMS and requires large amounts of computing resources and storage space.

---

> ### Author Response · Authors · 2024-11-17
> **Response to Reviewer pgv7 (3)**
>
> **Following to A3.3**:
> c. The RME-module with random masking designed in our DERI is used to enhance the global feature of the ECG signals. Compared with random dropout used in MERL, which removes a fraction of the neurons (units) or edges in a network layer during training, our RME-module uses two different random masks to mask the local representation of part of the ECG and introduces an attentional mechanism to extract global features from the masked ECG representation. By aligning these two independent masked global representations, our model can more efficiently extract the global features of the ECG for further cross-modal alignment and reconstruction.
>
> Comprehensive experiments on various downstream datasets are conducted to evaluate the proposed DERI method and experimental results have verified the great performance of our method.
>
> **Q3.4**: For instance, two of the losses used in the paper are almost identical to the ones used in MERL (CLIP loss and mask loss); the paper merely describes them in a different way. The report generation method is also quite similar to many multimodal approaches, such as the BLIP method, and does not represent true innovation. Moreover, these papers have not been cited.
>
> **A3.4**: On the one hand, CLIP loss has shown its great performance in multi-modal learning, such as Ref. [A], Ref. [B], and Ref. [C]. It is widely used for cross-modal representation learning, and this is the reason why we use this loss in our DERI. On the other hand, the main innovations of our method, which can be seen in A3.3, do not focus on the innovation of the loss function. BLIP also provides an effective method to generate text based on visual encoding, which is also widely used in clinical report generation such as R2Gen [D] and HERGen [E], we have cited these two great work in our articles.
>
> [A] Hafner M, Katsantoni M, Köster T, et al. CLIP and complementary methods[J]. Nature Reviews Methods Primers, 2021, 1(1): 1-23.
>
> [B] Zhang, R., Guo, Z., Zhang, W., Li, K., Miao, X., Cui, B., ... & Li, H. (2022). Pointclip: Point cloud understanding by clip. In Proceedings of the IEEE/CVF conference on computer vision and pattern recognition (pp. 8552-8562).
>
> [C] Fan, L., Krishnan, D., Isola, P., Katabi, D., & Tian, Y. (2024). Improving clip training with language rewrites. Advances in Neural Information Processing Systems, 36.
>
> [D] Chen, Z., Song, Y., Chang, T. H., & Wan, X. (2020). Generating radiology reports via memory-driven transformer. arXiv preprint arXiv:2010.16056.
>
> [F] Wang, F., Du, S., & Yu, L. (2024). HERGen: Elevating Radiology Report Generation with Longitudinal Data. arXiv preprint arXiv:2407.15158.
>
> We hope our response has addressed all concerns. We would greatly appreciate any further constructive comments or discussions.

---

### Official Review · Reviewer_pYTw · 2024-11-03

**Soundness:** 2
**Presentation:** 3
**Contribution:** 2
**Rating:** 3
**Confidence:** 5

**Summary:**

The study introduces DERI (Deep ECG-Report Interaction), a framework designed for cross-modal representation learning from ECG signals and clinical reports to improve the clinical relevance of ECG representations. Traditional ECG self-supervised methods focus on single-modal generative or contrastive learning but fail to capture the deep clinical semantics. DERI addresses these limitations by integrating ECG and report data more deeply.

**Strengths:**

- Deep Cross-Modal Interaction: Unlike previous methods that use shallow alignment between ECG and report features, DERI implements multiple alignments and feature reconstructions, fostering deeper interactions between ECG signals and report data. This approach strengthens the representation's ability to capture the clinical context, enhancing both accuracy and relevance for diagnosis.
- Potential for Broader Clinical Integration: DERI’s architecture, designed to integrate additional data types like electronic medical records (EMRs), positions it well for broader application in clinical settings. This flexibility could make DERI a powerful tool for multi-modal clinical analysis in the future.

**Weaknesses:**

In contrast to other modalities such as chest X-rays and pathological diagnoses, electrocardiogram reports have been mainly produced mechanically by diagnostic equipment for many years. Therefore, this study is more likely to be learning of waveform data and its correct labels, rather than two-modal learning of waveform data and its interpretation using natural language.
The scope of this study may be narrow than the general interest of ICLR main conference.

**Questions:**

Is it possible to disclose what the electrocardiogram reports used as training data in this study are specifically like? Was it verified how diverse the content of these reports is in terms of natural language?

---

> ### Author Response · Authors · 2024-11-17
> **Response to Reviewer pYTw**
>
> We appreciate your review and the positive comments regarding our paper. We would like to respond to your comments as follows.
>
> **Q2.1**: In contrast to other modalities such as chest X-rays and pathological diagnoses, electrocardiogram reports have been mainly produced mechanically by diagnostic equipment for many years. Therefore, this study is more likely to be learning of waveform data and its correct labels, rather than two-modal learning of waveform data and its interpretation using natural language. The scope of this study may be narrower than the general interest of the ICLR main conference.
>
> **A2.1**: Our work proposes to learn an effective representation of ECG signals with the help of diagnostic reports. During the pre-training process, we conducted Deep ECG-report Interaction to train the model to learn effective representation with more clinical cardiac information. After that, the model can be used to learn a representation of ECG signals without report. ECG classification and report generation are the downstream tasks we conduct to verify the ability of our DERI to learn ECG representation. Therefore, our method is a total representation learning model that fits perfectly with the scope of the International Conference on Learning Representations.
>
> **Q2.2**: Is it possible to disclose what the electrocardiogram reports used as training data in this study are specifically like? Was it verified how diverse the content of these reports is in terms of natural language?
>
> **A2.2**: Actually, we have provided some example samples of the diagnostic reports used as training data in the supplement, including short reports, medium reports, and long reports, which are also shown below:
>
> **Short report**: sinus bradycardia. prolonged qt interval. borderline ecg.\
>
> **Medium report**: sinus rhythm. poor r wave progression - probable normal variant. inferior st-t changes may be due to myocardial ischemia.
>
> **Long report**: sinus bradycardia. prolonged qt interval. possible anterior infarct - age undetermined. lateral t wave changes may be due to myocardial ischemia. abnormal ecg.
>
> To verify how diverse the content of these reports is in terms of natural language, we conduct a statistical analysis on the reports of MIMIC-ECG, which calculated the counts of each word on all reports, and rare word count, TTR and Herdan’s C of each report. Since there are 771,693 reports and the average sentence has 14 words, we consider rare words that appear no more than 0.1% of the total number of reports (771). Based on the rare word count, we calculate the TTR and Herdan’s C of each report as below:
>
>  $TTR = \frac{Rare Word Count}{Report Word Count}$
>
> $TTR = \frac{log(Rare Word Count)}{log(Report Word Count)}$
>
> Here, TTR is the ratio of the number of rare words (types) to the total number of words (order), reflecting the diversity of the text. A higher TTR indicates higher lexical diversity and less lexical repetition. Herdan's C reflects the lexical diversity of a text by calculating the logarithmic ratio of the number of rare words in the text to the total number of words. The results are shown below:
> |        | Word counts | Rare Word Count | TTR (%) | Herdan's C  |
> |--------|-------------|-----------------|---------|-------------|
> | Max    | 650142.00   | 18.00           | 83.33   | 0.90        |
> | Min    | 1.00        | 0.00            | 0.00    | 0.00        |
> | Median | 329.00      | 0.00            | 0.00    | 0.00        |
> | Mean   | 12990.21    | 0.09            | 0.52    | 0.01        |
> | Std.   | 48132.65    | 0.50            | 3.34    | 0.05        |
>
> We also find that there are 818 different words used in these reports while 486 words are regarded as rare words. Moreover, there are 40 words that appear only once in all reports while 110 words that appear less than 10 times in all reports. However, our distribution shift experiments show that our DERI method will not be limited by the scope and the bias of the report used in pre-training since it performs well for samples without reports.
>
> We hope our response has addressed all concerns. We would greatly appreciate any further constructive comments or discussions.

---

### Official Review · Reviewer_PSAx · 2024-11-03

**Soundness:** 3
**Presentation:** 2
**Contribution:** 3
**Rating:** 6
**Confidence:** 4

**Summary:**

The novel DERI approach enhances cross-modal representation learning by incorporating clinical report generation. The work extends the MERL approach [1], integrating multiple alignments and a report generative approach with a novel latent random masking module (RME). The novelty of the approach lies in not only aligning the ECG and report features but also decoding cross-modal features. The author demonstrated a performance improvement compared to other SOTA approaches, verified through supervised tasks on unseen data.

**Strengths:**

The work is a natural extension of MERL[1], with more accurate zero-shot classification and the possibility of automatic report generation. The zero-shot classification performance shows significant improvement from [1]. The cross-modal decoders allow the additional capability of automatic report generation utilizing GPT-2 architecture.

**Weaknesses:**

The paper needs rephrasing to improve clarity and readability, especially in the methodology section. The training approach is not applicable to unlabelled ECG in the usual context and necessitates the availability of accurate diagnostic reports by a cardiologist. The performance is related to the quality, distribution, and context of these reports and may not extend to novel tasks outside the scope of diagnostic reports.

**Questions:**

Methodology

Does incorporating textual reports in pre-training have an element of supervision?

Does the pre-training necessitate diagnostic reports for ECGs or can it also utilize ECGs when the reports are not available?

Are the reports automatically generated or written by cardiologists?

The strength of the self-supervised pretraining approach is learning general features while incorporating textual reports limits the features to the scope of the reports and thus might limit the capabilities for future tasks outside the information provided in the reports. Can the author demonstrate that the learned features are not limited by the scope and bias of the reports?

Does the alignment loss in Equation 1 accommodate the situation where multiple ECGs have similar text reports?

How does random masking differ from random dropout?

Is the performance for other approaches evaluated by the author or the original work since most models require a hyperparameter optimization for best performance?

General comments

Page 1 line 29: rephrase “clinical cardiac conditions classification”.

Page 2 lines (61-72): “Specifically, the ECG signal and …. as follows.” Please rephrase.

Page 2 line 78: What is meant by “which can provide clinical semantics visually”?

Page 2 line 91: “temporal and spatial correlation ship of ECG signals”. to “temporal and spatial correlations of ECG signals”.

Page 2 line 140: Details and references for the text encoder and masking/reconstruction are missing in the methodology section.

Page 4 line (164-194): Please provide references if equations 1 to 5 are derived from existing literature and indicate where there are novel concepts.

Page 4 line (202-207): “We introduced” to “We introduce”

Page 4 line (202-206): “Considering that the textual modality … order to provide more textual features”. Please rephrase to improve clarity and avoid very long sentences.

Page 4 lines (206-208): “After completing …. corresponding report text.” Please rephrase

Page 4 sec 3.2: What is meant by the decoded text and ECGs? Are these the reconstructions of the feature encoding or the corresponding aligned text? If encoding then how is it combined in the mixed-modal representation if the dimensions are different?

Page 7 lines (360-362): “The experimental results …. for classification”. Not clear please rephrase.

Figures: Figure captions need to be improved.

Figure 1: Please explain the figure adequately in the caption.

Tables: Table captions should include the supervised task and the metric under observation.

Table 7: Please change DERL to DERI.


References:

[1] Che Liu, Zhongwei Wan, Cheng Ouyang, Anand Shah, Wenjia Bai, and Rossella Arcucci. Zero-shot ecg classification with multimodal learning and test-time clinical knowledge enhancement. arXiv preprint arXiv:2403.06659, 2024.

---

> ### Author Response · Authors · 2024-11-17
> **Response to Reviewer PSAx (1)**
>
> We appreciate your review and the positive comments regarding our paper. We would like to respond to your comments as follows.
>
> **Q1.1**: The paper needs rephrasing to improve clarity and readability, especially in the methodology section.
>
> **A1.1**: We will improve the clarity and readability of our paper, check and correct all grammatical errors, and improve where the description is vague. We use the expression "provide clinical semantics visually" because clinical reporting information can directly provide clinical semantics for ECG interpretation. "visually" is used to show the meaning of "directly", and we will revise it in our revised manuscript.
>
> Details and references for the text encoder and masking/reconstruction will also be provided. Specifically, for the encoders used for ECG and reports, we adopt a random initialized 1D ResNet18 and the pre-trained MedCPT Query Encoder. MedCPT Query Encoder can generate embeddings of biomedical texts for semantic search (dense retrieval), which is suitable for our work. For cross-modal mutual reconstruction, we introduce two transformer decoders to reconstruct the representation of one modality with the other modality (use ECG embedding to reconstruct report embedding and use report embedding to generate ECG embedding). Detailed parameter settings for the decoder are provided in Section 4.2 lines 312-314. For random masking in RME-module, given a representation $x \in R^{b,n,c}$, we first generate a random Gaussian noise $m \in R^{b,n}$. The noise of each sample is sorted in ascending order to get the index of each sample. We choose the $k$ (decided by masking ratio) element indexes with the highest noise and the least noise as the masked feature indexes respectively, to ensure that the two masked representations are different.
>
> The loss function provided in equations 1 to 5 in our method adopts the CLIP loss (I), which has been widely used in multi-modal contrastive learning. These equations are used to clearly illustrate how we conduct cross-modal representation alignment.
>
> For Page 4 sec 3.2, cross-modal decoders are used to reconstruct the representation of each modal with the other modal, in other words, we reconstruct ECG encoding and text encoding as shown in Figure 1. Linear projectors $P_{e}$ and $P_{t}$ are used to map them into an alignment space of the same dimension for obtaining mixed-modal representation.
>
> Page 7 lines (360-362) are rephrased as “The experimental results of linear probing are provided in Table 1. The ‘Random Init’ in Table 1 represents using the model structure of our proposed DERI to obtain the ECG-specific mix encoding without pre-training. The model is trained on the downstream dataset in a fully supervised manner for ECG classification.”
>
> We also improve the captions of figures and tables in our paper for better clarity.
>
> **Q1.2**: The training approach does not apply to unlabeled ECG in the usual context and necessitates the availability of accurate diagnostic reports by a cardiologist.
>
> **A1.2**: Diagnostic reports are used in the pre-training stage of our method. We proposed a novel framework containing ECG-Report Multiple Alignment and Cross-Modal Mutual Reconstruction to enable the model to learn a more effective representation of the ECG signal. The reports provide a clinical description of the ECG signal, which contains direct high-level semantics. After pre-training, our model can effectively learn the representation of ECG signals which contain high-level semantic insights without diagnostic reports and labels. This is also why DERI has obtained significant improvement over the baselines on public datasets for ECG classification. Experiments on report generation also show that our DERI can learn more high-level semantic information than MERL, which only aligns ECG features with report features in a relatively shallow way.
>
> **Q1.3**: Does incorporating textual reports in pre-training have an element of supervision?
>
> **A1.3**: We believe that in a sense the inclusion of text reports in pre-training does have some element of supervision. The text reports provide additional semantic information to the ECG signal. The clinical information contained in each report actually guides the ECG signal as it represents the professional interpretation of the physician. This means that the model captures specific semantics by aligning the ECG signals with the corresponding reports, and this alignment process is equivalent to some form of supervised learning as the textual content helps the model to identify key features in the signals. However, the doctor's diagnostic report is not exactly equivalent to the category labeling of the ECG signal but is more of an interpretation of the ECG waveform. However specific diseases tend to have multiple waveform states and different diseases can have the same waveform. Therefore we do not consider the introduction of diagnostic reports for representation learning to be supervised learning.

---

> > ### Author Response · Authors · 2024-11-17
> > **Response to Reviewer PSAx (2)**
> >
> > **Q1.4**: Does the pre-training necessitate diagnostic reports for ECGs or can it also utilize ECGs when the reports are not available?
> >
> > **A1.4**: Our method is proposed to conduct deep ECG-Report Interaction, which can learn representation with more effective cardiac information. Therefore, diagnostic reports are necessary for pre-training. Without reports, our pre-training method cannot conduct ECG-Report Multiple Alignment and Cross-Modal Mutual Reconstruction. However, after pre-training, our method can utilize ECGs only in the downstream tasks including classification and report generation.
> >
> > **Q1.5**: Are the reports automatically generated or written by cardiologists?
> >
> > **A1.5**: For the MIMIC-ECG dataset used for pre-training, the reports are reported by a cardiologist according to Ref. [1]. Moreover, we conduct more experiments about ECG-report generation on the PTB-XL dataset, which is not used in pre-training. For the PTB-XL dataset, the reports are generated by the cardiologist or automatically interpreted by the ECG device according to Ref. [2]. Experimental results are shown in the table below:
> >
> > |               |     Method                | BLEU-1 | BLEU-2 | BLEU-3 | BLEU-4 | ROUGE-L  |
> > |---------------|--------------------|--------|--------|--------|--------|----------|
> > | Our Framework | MERL-DisGPT2       | 48.9   | 44.4   | 41.2   | 37.4   | 55.4     |
> > |               | DERI-DisGPT2       | **58.6**  |**54.8**   | **51.9**   |**48.6**   | **64.2**     |
> > |               | DERI-Align-DisGPT2 | 54.1   | 49.8   | 46.6   | 43.2   | 60.1     |
> > |---------------|--------------------|--------|--------|--------|--------|----------|
> > | Ref [1]       | GPT2-Medium        | 32.9   | 27.8   | 25.4   | 23.2   | 39.1     |
> > |               | GPT2-Large         | 43.7   | 39.5   | 35.5   | 32.0   | 48.1     |
> > |               | GPT-Neo            | 47.4   | 44.9   | 39.8   | 37.3   | 48.6     |
> > |               | GPT-NeoX           | 46.9   | 45.3   | 41.7   | 39.9   | 55.3     |
> > |               | GPT-J              | 48.5   | 45.2   | 42.8   | 40.5   | 55.0     |
> > |               | BLOOM              | 49.1   | 46.2   | 42.7   | 41.5   | 58.0     |
> > |               | OPT                | 50.2   | 47.7   | 43.1   | 41.8   | 56.8     |
> > |               | LLaMA-1            | 51.4   | 48.5   | 46.5   | 43.0   | 58.8     |
> > |               | Mistral            | 48.6   | 47.5   | 44.6   | 42.1   | 59.1     |
> > |               | LLaMA-2†           | 51.5   | 48.4   | 46.9   | 43.9   | 59.4     |
> > |               | Mistral-Instruct†  | 50.1   | 48.1   | 45.7   | 42.5   | 59.2     |
> > |---------------|--------------------|--------|--------|--------|--------|----------|
> > | Ref [4]       | PTB-XL             | 6.5    | -      | -      | 0.9    | 25.6     |
> > |               | ECG-Chat           | 15.9   | -      | -      | 2.3    | 23.9     |
> > |               | ECG-Chat-DDP       | 32.3   | -      | -      | 11.2   | 29.9     |
> >
> > [1] Gow, B., Pollard, T., Nathanson, L. A., Johnson, A., Moody, B., Fernandes, C., Greenbaum, N., Waks, J. W., Eslami, P., Carbonati, T., Chaudhari, A., Herbst, E., Moukheiber, D., Berkowitz, S., Mark, R., & Horng, S. (2023). MIMIC-IV-ECG: Diagnostic Electrocardiogram Matched Subset (version 1.0). PhysioNet. https://doi.org/10.13026/4nqg-sb35
> >
> > [2] Wagner, P., Strodthoff, N., Bousseljot, R., Samek, W., & Schaeffter, T. (2020). PTB-XL, a large publicly available electrocardiography dataset (version 1.0.1). PhysioNet. https://doi.org/10.13026/x4td-x982.
> >
> > **Q1.6**: The strength of the self-supervised pretraining approach is learning general features while incorporating textual reports limits the features to the scope of the reports and thus might limit the capabilities for future tasks outside the information provided in the reports. Can the author demonstrate that the learned features are not limited by the scope and bias of the reports?
> >
> > **A1.6**: Incorporating textual reports into pre-training may result in the features learned by the model being limited to the scope and preferences of the report, which may limit the model's performance on tasks outside the scope of the report. Therefore, we conduct distribution shift experiments on the other three datasets without reports, which can effectively validate the learning ability of representation learning models for different data domains as Table 2 in our article. These datasets are from different healthcare organizations than the pre-training dataset and can effectively reveal whether the model has developed a preference for certain report content or specific data sources. The stable cross-domain performance of DERI in the experimental results suggests that the model has better generalization capabilities that are not limited to the information in the reports.

---

> > > ### Author Response · Authors · 2024-11-17
> > > **Response to Reviewer PSAx (3)**
> > >
> > > **Q1.7**: Does the alignment loss in Equation 1 accommodate the situation where multiple ECGs have similar text reports?
> > >
> > > **A1.7**: The alignment loss in Eq.1 regards each ECG signal with its corresponding report as a positive sample pair while multiple ECGs that have similar text reports will be regarded as negative pairs. This is because although some ECG signals have partially similar reports, their totality may represent different clinical presentations, and considering these signals as pairs of positive samples tends to make the model constrained by these overlapping reported elements in learning the representation.
> > >
> > > **Q1.8**: How does random masking differ from random dropout?
> > >
> > > **{A1.8**: In random masking, specific parts of the input signal embedding are masked. Specifically, after the Resnet encoder, the ECG embedding contains local features (R^{b,n,c}). We conduct random masking on the dimension n to mask several local features and then use the RME-module to enable the model to learn an effective global feature with fewer local features. Masking encourages the model to learn context-aware representations, focusing on understanding relationships within the input. Random dropout removes (i.e., zeroes out) a fraction of the neurons (units) or edges in a network layer during training, but it does not apply this to the input itself. In dropout, neurons are randomly dropped independently at each forward pass. The RME-module designed in our DERI is used to enhance the global feature of the ECG signals, so we adopt random masking rather than a random dropout. Our experimental result in the ablation study supports the effect of random masking.
> > >
> > > **Q1.9**: Is the performance for other approaches evaluated by the author or the original work since most models require a hyperparameter optimization for best performance?
> > >
> > > **A1.9**: The performance results for other approaches refer to the existing work "Zero-Shot ECG Classification with Multimodal Learning and Test-time Clinical Knowledge Enhancement (MERL)". However, for report generation, we adopt the same setting for MERL and our DERI as a comparison.
> > >
> > > We hope our response has addressed all concerns. We would greatly appreciate any further constructive comments or discussions.

---

### Note · Authors · 2024-11-26

**Comment:**

Paper withdrawn during the rebuttal period upon request by author(s).

**Withdrawal Confirmation:**

I have read and agree with the venue's withdrawal policy on behalf of myself and my co-authors.